# C5aR1 antagonism suppresses inflammatory glial responses and alters cellular signaling in an Alzheimer's disease mouse model

Nicole D. Schartz [1], Heidi Y. Liang[2], Klebea Carvalho [2], Shu-Hui Chu [1], Adrian Mendoza-Arvilla[1], Tiffany J. Petrisko[1], Angela Gomez-Arboledas [1], Ali Mortazavi [2] & Andrea J. Tenner [1,3,4] ✉

Alzheimer's disease (AD) is the leading cause of dementia in older adults, and the need for effective, sustainable therapeutic targets is imperative. The complement pathway has been proposed as a therapeutic target. C5aR1 inhibition reduces plaque load, gliosis, and memory deficits in animal models, however, the cellular bases underlying this neuroprotection were unclear. Here, we show that the C5aR1 antagonist PMX205 improves outcomes in the Arctic48 mouse model of AD. A combination of single cell and single nucleus RNA-seq analysis of hippocampi derived from males and females identified neurotoxic disease-associated microglia clusters in Arctic mice that are C5aR1-dependent, while microglial genes associated with synapse organization and transmission and learning were overrepresented in PMX205-treated mice. PMX205 also reduced neurotoxic astrocyte gene expression, but clusters associated with protective responses to injury were unchanged. C5aR1 inhibition promoted mRNA-predicted signaling pathways between brain cell types associated with cell growth and repair, while suppressing inflammatory pathways. Finally, although hippocampal plaque load was unaffected, PMX205 prevented deficits in short-term memory in female Arctic mice. In conclusion, C5aR1 inhibition prevents cognitive loss, limits detrimental glial polarization while permitting neuroprotective responses, as well as leaving most protective functions of complement intact, making C5aR1 antagonism an attractive therapeutic strategy for AD.

Alzheimer's disease (AD) is the most common form of dementia in the elderly with an estimated 6.5 million people currently diagnosed in the US and with an expected doubling by 2060[1]. AD pathology is characterized by the accumulation of extracellular amyloid beta (Aβ) plaques, hyperphosphorylated tau, and neuronal loss ultimately resulting in cognitive decline[2]. Although Aβ deposition and tau phosphorylation are the hallmarks of AD, evidence suggests that inflammation triggered by these events and/or occurring prior to pathology contribute to onset and acceleration of functional loss and thus may be a more appropriate, broadly effective therapeutic target to prevent disease progression[3]. Indeed while AD subtypes suggesting diverse etiology are evident[4,5], neuroinflammation and neurodegeneration are commonly evident with the loss of cognition[6,7].

---

[1]Department of Molecular Biology & Biochemistry, University of California, Irvine, Irvine, CA, USA. [2]Department of Developmental & Cell Biology, University of California, Irvine, Irvine, CA, USA. [3]Department of Neurobiology and Behavior, University of California, Irvine, Irvine, CA, USA. [4]Department of Pathology and Laboratory Medicine, University of California, Irvine, School of Medicine, Irvine, CA, USA. ✉e-mail: atenner@uci.edu

Complement activation is critical for the rapid recognition and clearance of pathogens, apoptotic cells, and cellular debris. Activation of the complement system in the CNS promotes opsonization via C3b/iC3b, myeloid cell recruitment via the production of C3a and C5a, and cell lysis via the membrane attack complex (MAC)[8]. Classical complement pathway component C1q has been largely shown to be neuroprotective in in vitro systems by promoting the clearance of apoptotic cells, modulating subsequent inflammatory cytokine production, and directly enhancing survival of neurons[9,10]. However, under pathological conditions such as AD, induced expression of additional complement components in brain enable the activation of the entire complement cascade (reviewed in refs. 11,12, including the cleavage of C5 into C5a and C5b fragments and thus resulting in induction of inflammation as well as neuronal damage. In human AD and mouse models of AD, complement components C1q, C3b, and C4b co-localize with fibrillar Aβ plaques (reviewed in refs. 3,13) and C5b-9 has also been detected in areas of plaques[14,15] providing evidence of amyloid-associated complement pathway activation.

The primary C5a receptor (C5aR1, aka CD88) is expressed at low levels in microglia in the CNS under physiological conditions, but under pathological conditions it can be highly upregulated and may be expressed by other cell types such as neurons and endothelial cells[16,17]. In mouse models of AD, C5aR1 is most highly expressed in plaque-associated microglia[18], and genetic ablation or inhibition of C5aR1 or C5a restores cognitive performance and reduces neuroinflammation[19–22]. C5aR1 exerts its effects via cAMP and ERK1/2 signaling as well as recruitment of β-arrestin 2[23,24]. C5a-C5aR1 engagement results in potent pro-inflammatory responses via MAPK signaling and NFκb activation, resulting in generation and secretion of pro-inflammatory mediators, and is known to synergize with innate immune pathogen recognition receptors to enhance pro-inflammatory cytokine responses[25–27]. Thus, the neuroprotective effect resulting from C5aR1 inhibition in AD models may include direct effects on cellular production of inflammatory and neurotoxic products as well as the prevention of downstream intercellular signaling pathways involving microglia and/or other cell types including neurons, astrocytes, oligodendrocytes, or endothelial cells that subsequently result in generation of neurotoxic responses.

PMX205 is a cyclic hexapeptide noncompetitive inhibitor of C5aR1[28]. With high oral bioavailability, low accumulation in the blood, brain, or spinal cord, and efficacy at crossing the blood brain barrier, it is a viable candidate for chronic treatment to block C5a-C5aR1 signaling in the brain[28]. PMX205 treatment has been reported to exert neuroprotective effects in models of neurodegeneration, including experimental autoimmune encephalomyelitis[29], amyotrophic lateral sclerosis[30], and spinal cord injury[31]. We have previously reported that PMX205 treatment initiated at the onset of amyloid deposition in the Tg2576 mouse model of AD reduces amyloid plaque accumulation and dystrophic neurites, alters gene expression in microglia, and improves cognitive performance[21,22]. In the present study, we treated the aggressive Arctic48 mouse model of AD with PMX205 after substantial amyloid plaques were already present to determine the protective effects of C5aR1 inhibition in advanced amyloid pathology. We tested spatial memory performance with the Y maze and assessed hippocampal single cell (SC) microglial or single nucleus (SN) preparations by RNA-seq. As expected, Arctic mice had a robust upregulation of inflammatory microglial gene expression relative to wild-type mice. Importantly, in the Arctic hippocampus, PMX205 treatment reduced inflammatory microglial gene expression while maintaining the beneficial responses to injury and promoted intercellular protective signaling pathways between different brain cell types including transforming growth factor-β (TGF β), bone morphogenetic protein (BMP), and fibroblast growth factor (FGF). In addition, short-term spatial memory deficits observed in Arctic females at 10 months of age were prevented by treatment with PMX205.

## Results

### Treatment with PMX205 suppresses complement gene transcription in Arctic mice

Mice were treated with PMX205 (20 μg/ml) or not in drinking water from 7.5 to 10 months of age when plaque accumulation in the hippocampus rises and plateaus[20] (Fig. 1a). Mice did not lose weight during the course of the treatment (Supplementary Fig. 1a, b) and consumed an average of 6-10 mL of water or PMX205 in water per day (Supplementary Fig. 1c, d), resulting in an average dose of 4.76 mg/kg and 5.36 mg/kg of PMX205 per day for WT and Arctic (Arc) mice, respectively (Supplementary Fig. 1e), consistent with doses in previous studies[22]. WT males had a lower relative average daily dose of PMX205 due to the higher body weight of WT males (Supplementary Fig. 1f).

Transcripts from single cell microglia and single nucleus libraries derived from hippocampi were combined and clusters were identified with Seurat. Of the 42 Seurat clusters containing 54,157 cells/nuclei identified (Figs. 1b, c, and 2a), 26 were identified as neurons, 5 as microglia, and 3 as astrocytes (Fig. 1b, d and Supplementary Data 1). Among the single nucleus samples, neurons made up the majority of cell types sequenced from each of the treatment groups (Fig. 2b). While most clusters were derived from single nucleus populations, microglial clusters were derived mostly from single cell isolation although one microglial cluster was exclusively derived from single nucleus (Fig. 1d). Interestingly, some microglial clusters (13, 16 and 23) from SN- and SC-RNA-seq were segregated between nuclear and whole cell locations (Fig. 1c, d), which is likely the result of compartment-dependent differences in mRNA half-life and/or populations[32]. We also identified 2 clusters of oligodendrocytes and 2 of oligodendrocyte progenitor cells (OPCs), as well as single clusters of endothelial cells, pericytes, and two "mixed" clusters that are likely a mixture of astrocytes and microglia (and thus were excluded from subsequent analyses) (Fig. 1d).

Specific clusters were overrepresented in the Arc samples compared to WT, including microglial clusters 13 and 15, and astrocyte cluster 10 (Fig. 1e). Because we used a treatment to modulate complement activation-mediated signaling, we further explored the relative complement gene expression in each cell cluster and genotype/ treatment group (Fig. 2, Supplementary Fig. 2a). Neuronal clusters expressed relatively high levels of complement regulatory genes *Csmd1* and *Csmd2* (irrespective of genotype), and astrocytes had high transcript levels of *C4b*, *C3* and *Serping1* (C1 Inhibitor) expressed predominantly in Arc mice (cluster 10) (Figs. 1f, and 2c). Microglial clusters, again particularly those unique to Arctic mice, expressed *C1qa*, *C1qb*, *C1qc*, *Cfh* (Factor H), *Itgam* (CD11b), *Itgb2* (CD18), *Itgax* (CD11c), *C5ar1*, *C5ar2*, and *C3ar1* (Figs. 2c and 1f). When only looking at SN samples, neuronal, endothelial, and OPC expression of complement genes was unchanged, while complement genes in astrocytes and microglia were upregulated in Arc compared to WT (Fig. 2c). To discriminate in a more focused analysis on the C5aR1-dependent and -independent gene expression, microglia, and astrocyte data were separately re-clustered.

### PMX205 treatment results in downregulation of inflammatory and neurotoxic microglia genes

We subclustered the SC and SN microglia clusters without the other brain cell types to identify 11 clusters (Fig. 3a). Based on very low expression levels of known microglial genes such as *Csf1r*, *Hexb*, *Cx3cr1*, we determined that cluster M11 had limited microglia, and it was excluded from further analysis (Supplementary Fig. 2b). Comparisons of the proportional distributions of microglial clusters within each genotype/treatment group revealed clusters that were enriched in WT cells (cluster M5), clusters that were specific to Arctic cells regardless of PMX205 treatment (clusters M3 and M8), clusters that were highly enriched in Arctic cells but ablated with treatment of PMX205 (clusters M4, M6 and M9), and clusters that were enriched in

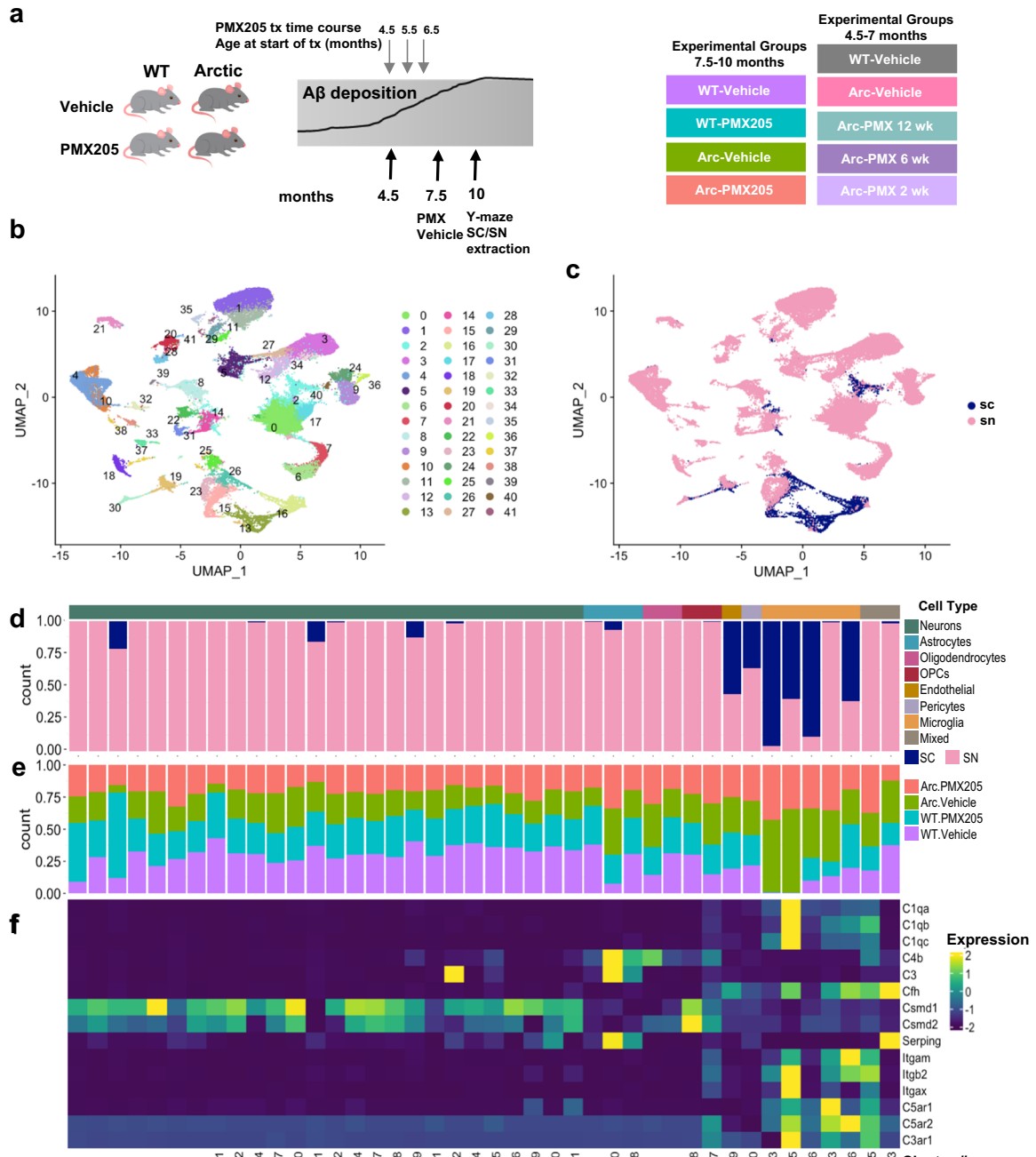

**Fig. 1 | Clustering of transcriptome from single nucleus and single cells in wild-type and Arctic mice. a** Diagrammatic representation of treatment paradigm for each of the experiments, including treatment from 4.5 to 7 months (top) and 7.5 to 10 months (bottom). **b** Seurat cluster identification of combined cells/nuclei transcriptome. **c** Seurat cluster identification of cells derived from single cells (SC) or single nucleus (SN) sequencing. **d** Proportion of cells in each cluster originating from SC or SN transcriptome. **e** Proportion of cells in each cluster by treatment/genotype. **f** Expression of complement pathway components, receptors or regulators by cell type.

cells upon PMX205 treatment (cluster M1) (Fig. 3b). Hierarchical clustering was used to segregate microglial clusters based on similarities of marker gene expression and then to assess proportions among SC/SN preps (Fig. 3c) and treatment groups (Fig. 3d).

Based on the current categorization that is widely used in the literature[33], we assessed relative expression of homeostatic, DAM1, and DAM2 genes by clusters as well as other genes of interest with SC and SN preparations grouped together (Fig. 3c–e) and with SC microglia analyzed separately (Fig. 3f, g). DAM1 gene transcripts (Cluster M4, Fig. 3c–e) are primarily found in the single cell, not the nuclear preparations. Interestingly, DAM1 genes were overexpressed in Arc

females compared to Arc males (Supplementary Fig. 3) that is consistent with an accelerated timeline to neuroinflammation in females. Notably, cluster M5, with high expression of homeostatic microglial genes, such as *Selplg* and *P2ry12* (Fig. 3e and Supplementary Data 2), was highly represented in WT microglia, regardless of PMX205 treatment (Fig. 3b). Clusters M3 (*Axl*) and M8 (*Itgax* and *Spp1*) were present only in the Arc mice regardless of PMX205 treatment and had low DAM1 gene expression. M8 had the highest relative expression of DAM2 genes (Fig. 3e). This suggests that the DAM2-induced responses to injury represented in clusters M3 and M8 were independent of C5aR1 signaling.

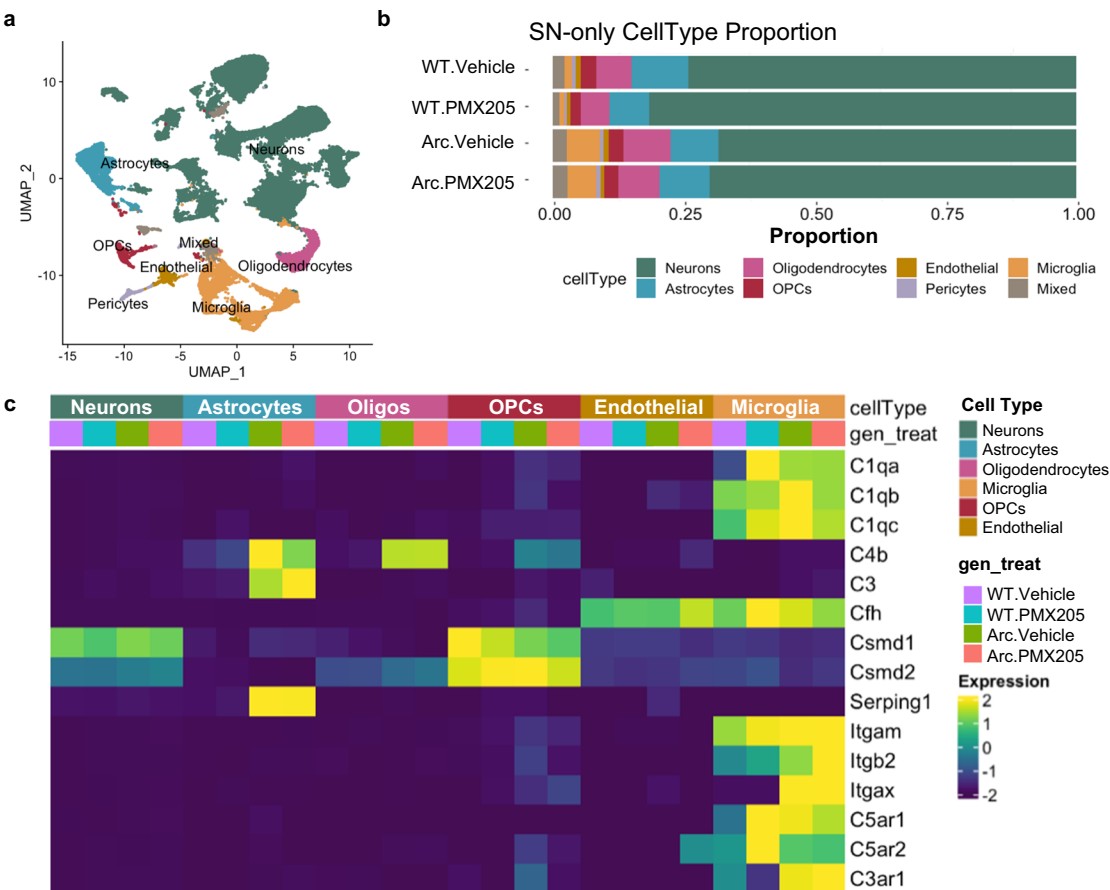

**Fig. 2 | Single cell and single nucleus RNA-Seq clustered by cell type reveals cell-specific complement gene expression.** Microglia or nuclei were isolated from hippocampi, fixed, and sequenced. **a** U-Map of all cell types with all treatment groups included (counts per cell type were: Neurons 37512, Astrocytes 4612, Oligodendrocytes 3478, Microglia 4891, OPCs 1259, Endothelial 898, Mixed 1017, Pericytes 490). **b** Proportion of isolated cell types in the single nucleus (SN) RNA-seq by genotype and treatment group. **c** Differential cell type-specific expression of complement pathway components and regulators of the complement system derived from single nucleus RNA-seq.

In contrast, clusters M4, M6, and M9 were highly induced in the Arc hippocampus, but only minimally detected in Arc-PMX205. Top genes of cluster M4 included *Apoe*, *C1qa*, *C1qb*, *C1qc*, *Tyrobp*, *Trem2*, *Lyz2* and *Cst7* (Fig. 3e, Supplementary Data 2). These DAM1 genes were highly expressed in Arc microglia compared to WT microglia but were not so highly induced in Arc-PMX205 microglia (Fig. 3g). GO analysis for cluster M4 identified pathways associated with synapse pruning, microglial cell activation, positive regulation of cell death, and regulation of myeloid cell differentiation (Supplementary Fig. 4a) suggesting that this population may be contributing to detrimental effects that are blocked by inhibition of C5aR1 signaling. Similarly, cluster M9 (with high expression of *C5ar1*) expressed disease-associated microglial genes *Inpp5d*, *Mertk*, and *Tgfbr2* (Supplementary Data 2). GO analysis of cluster M9 genes revealed pathways associated with regulation of lymphocyte activation and cytokine production (Supplementary Fig. 4b). Cluster M6 was differentiated from other clusters by only 16 genes (Supplementary Data 2). Among those genes were modulators of the immune system, *Lars2* that supports a higher metabolic state, *Cmss1* involved in translation, and *Cdk8* and *Cd44* that promote inflammation. Treatment with PMX205 completely suppresses those injury-induced inflammation-enhancing genes. In contrast, microglia cluster M2 made up a smaller proportion in Arc cells relative to WT but was largely rescued in Arc-PMX205 microglia. Genes found in microglial cluster M2 were associated with synapse organization, regulation of glutamatergic transmission, and maintenance of synapse structure (Supplementary Fig. 4c).

Finally, Cluster M1, which is dependent on PMX205 treatment regardless of genotype, contained genes associated with mRNA splicing and microtubule cytoskeleton organization and processes. Although this cluster was not defined by many genes, it indicates that lack of C5aR1 signaling results in improved glial structural integrity and neuroprotective functions. In summary, homeostatic genes were highly represented in WT-Veh and WT-PMX205 microglia, while DAM1 genes were highly upregulated in Arc-Veh cells. Treatment with PMX205 had a strong effect in reducing DAM1 gene expression, but less so on DAM2 expression. That is, C5aR1 is a driver of the detrimental inflammatory injury response to amyloid in the hippocampus and that inhibition in Arc mice enhances pathways associated with synaptic plasticity and learning while suppressing pathways associated with glial activation, synaptic loss, and cell death.

**PMX205 treatment reduces expression of some neurotoxic astrocyte genes**

Astrocyte clusters were also isolated and re-clustered for further exploration of astrocyte subpopulations. Twelve astrocyte (A) clusters were defined (Fig. 4a) and confirmed to contain astrocytes based on gene expression (Supplementary Fig. 2c). Of these, clusters A1 and A2 were dominant in WT cells, while cluster A3 was specific to Arc cells regardless of PMX205 treatment and cluster A7 was high in Arc cells and ablated with PMX205 treatment (Fig. 4b). In contrast, cluster A6 was depleted in Arc but rescued in Arc-PMX205.

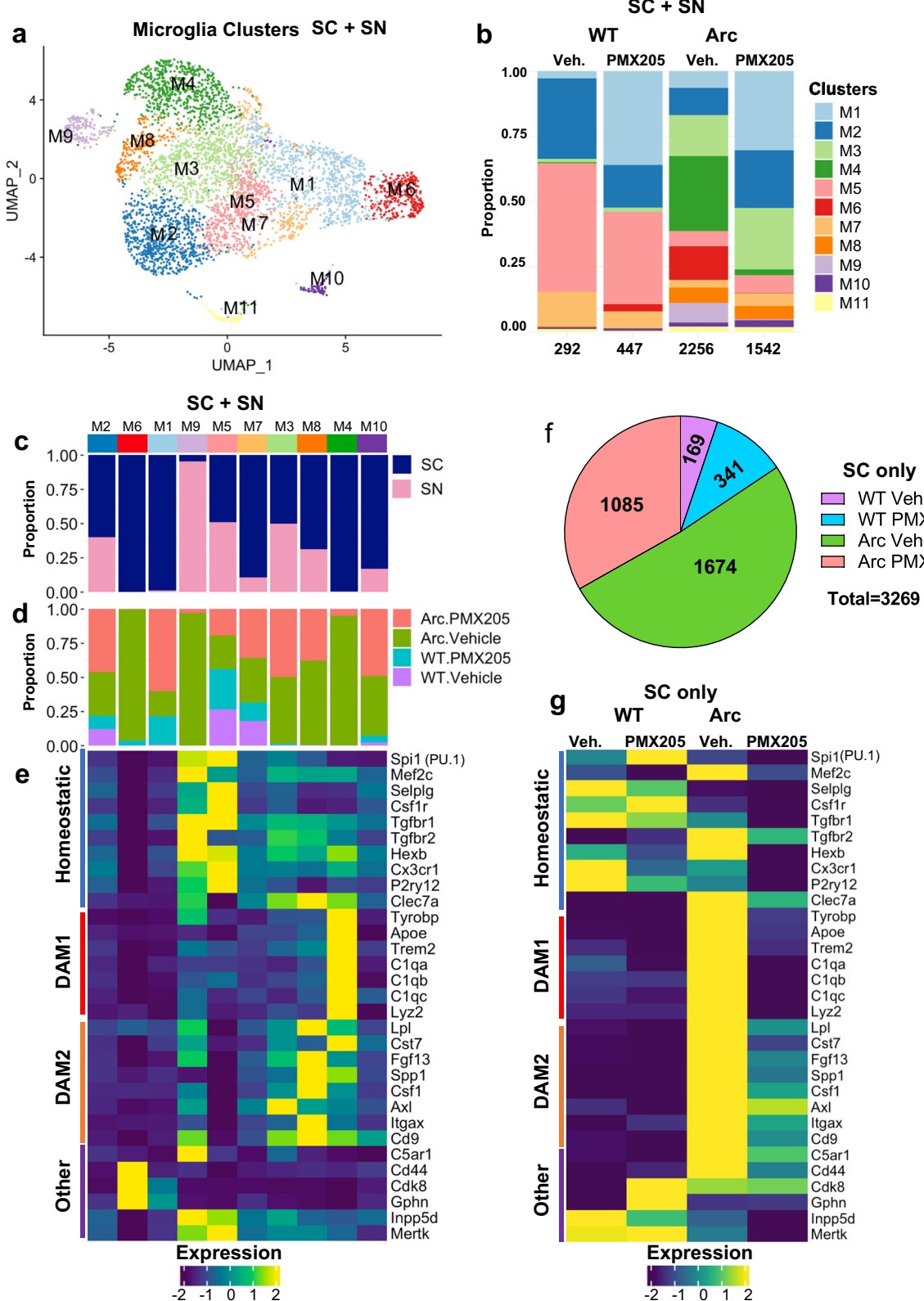

**Fig. 3 | DAM1 gene expression is suppressed in Arctic-PMX205 hippocampal microglia. a** Cells/nuclei identified as microglia were re-clustered separately. **b** Proportion of microglia (SC + SN) clusters in WT-Veh, WT-PMX, Arc-Veh, and Arc-PMX samples. **c** Proportion of cells in each cluster originating from SC or SN transcriptome. **d** Proportion of cells/nuclei in each cluster by treatment/genotype.

**e** Relative expression of genes representative of homeostatic microglia, DAM1, DAM2, or other genes of interest within the different microglial clusters. **f** Pie chart demonstrating proportion of SC microglia samples derived from different treatment groups. **g** Relative expression of homeostatic, DAM1, or DAM2 genes in different treatment groups, with SC data alone.

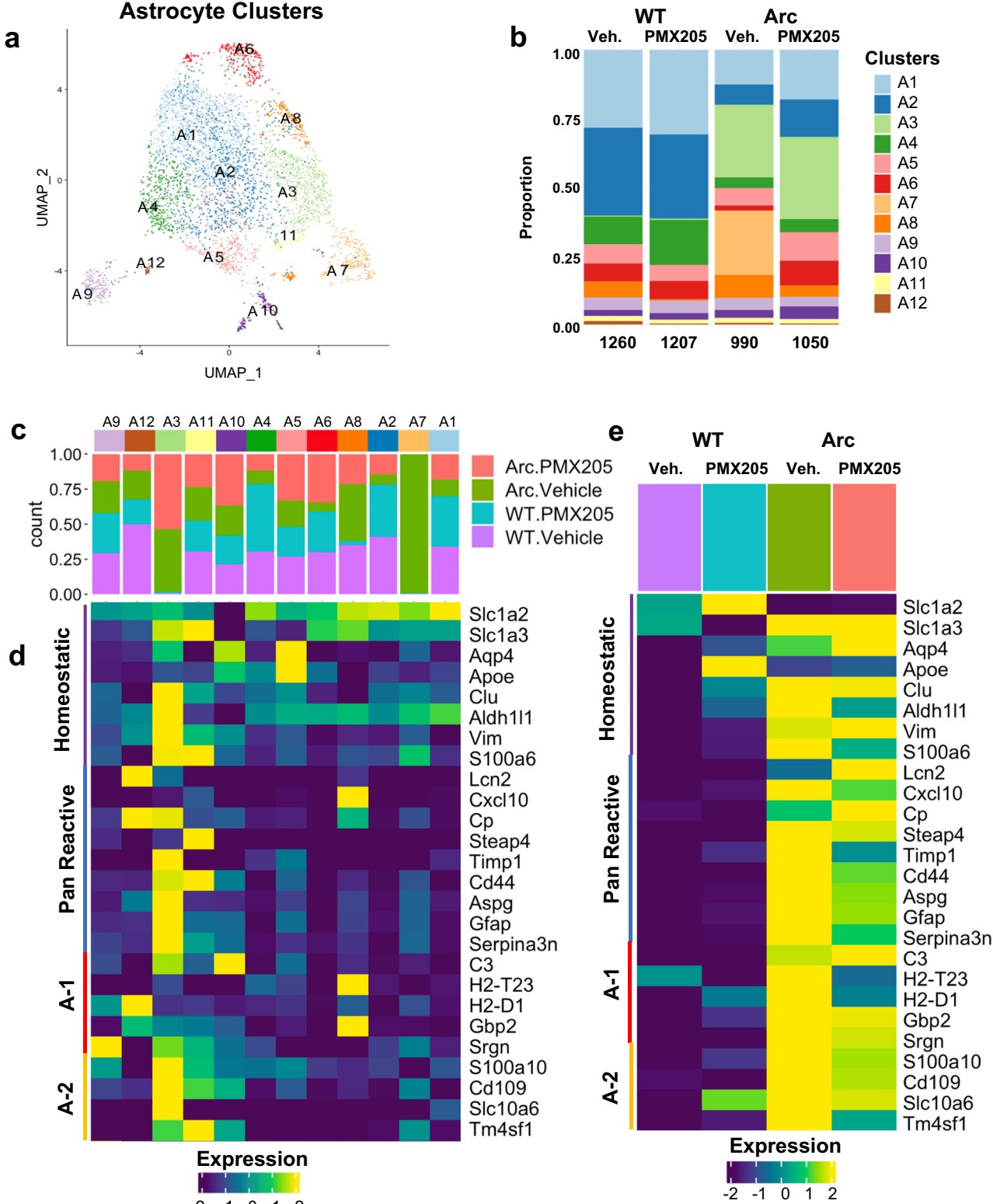

**Fig. 4 | Reactive Astrocyte gene expression is largely suppressed in Arctic-PMX205 hippocampus. a** Cells identified as astrocytes were re-clustered separately. **b** Proportion of astrocyte populations in WT-Veh, WT-PMX, Arc-Veh, and Arc-PMX samples. **c** Proportion of cells in each cluster by treatment/genotype. **d** Relative expression of genes representative of homeostatic astrocytes, pan-reactive, A-1 neurotoxic, or A-2 neuroprotective astrocytes within the different astrocyte clusters. **e** Relative expression of homeostatic, pan-reactive, A-1, or A-2 genes in the different treatment groups.

Cluster A3, which included *Gfap*, *C4b* and *Cd44* amongst its top genes (Supplementary Data 3), had GO terms associated with tyrosine kinase signaling pathways, nerve development, response to wounding, and JAK-STAT pathway (Supplementary Fig. 5a). Interestingly, cluster A3 was highly enriched for pan-reactive and A-2[34] reactive genes (Fig. 4d), thus supporting that the function of this population is associated with disease-mitigating response to inflammation and injury that is independent of C5aR1 signaling.

Cluster A6 was suppressed in Arc astrocytes and was rescued by PMX205 (Fig. 4b, c). Cluster A6 GO terms included axon guidance, protein processing, and neuron differentiation (Supplementary Fig. 5b). Interestingly, cluster A10, increased slightly in Arc but more so in PMX205-treated Arc, had GO terms including synaptic signaling and neuron projection development (Supplementary Fig. 5c)., Thus, enhancement of clusters A6 and A10 by PMX205 is consistent with a heightened phenotype in astrocytes when C5a-C5aR1 signaling is inhibited that results in support of neuronal functions in response to inflammatory injury.

Cluster A7, which was unique to Arc astrocytes and not found in Arc-PMX205 cells, was only defined by 10 genes including *Arhgef4*, *Rgs6*, and *Enox1* (Supplementary Fig. 5d), and had low relative expression of previously identified reactive astrocyte genes (Fig. 4d, Supplementary Data 3). Of these defining genes, *Arhgef4* (Rho guanine nucleotide exchange factor 4) has been reported to play a role in G protein-coupled receptor-mediated responses to extracellular stimuli[35]. *Rgs6* also encodes a protein associated with regulation of G protein signaling[36]. SNPs of *Rgs6* have been implicated in numerous psychiatric disorders[37] and its overexpression promotes inflammation and oxidative stress in spinal cord injury[38]. Lastly, *Enox1*, involved in plasma membrane electron transport pathways, is associated with memory deficits in patients with psychiatric conditions[39]. Thus, existing evidence supports that astrocyte cluster A7 contains a population of disease-enhancing astrocytes that promote inflammation and neuronal dysfunction in Arctic mice that is completely suppressed by inhibition of C5aR1 signaling, defining a previously undescribed disease-associated astrocyte that is induced only in the presence of C5aR1 signaling.

Astrocyte clusters were analyzed for genes associated with pan-reactive, A-1 neurotoxic, or A-2 astrocytes[34]. The WT-enriched clusters A1 and A2 had very low expression of reactive genes (Fig. 4d). In contrast, cluster A3, highly elevated in all Arctic samples regardless of PMX205 treatment as mentioned above, showed high expression of pan-reactive and A-2 astrocyte genes, demonstrating these as response to injury independent of C5aR1 signaling. Interestingly, pan-reactive markers *Lcn2* and *Cp* (ceruloplasmin) were more highly expressed in Arc-PMX205 astrocytes compared to Arc-Veh, while A-1 genes *H2-T23* and *H2-D1* were strongly downregulated in Arc-PMX205 astrocytes. Overall, while the previously identified pan-reactive, A-1, and A-2 genes were enhanced in Arc astrocytes, PMX205 treatment substantially decreased only A1 *H2-T23* and *H2-D1*, and muted A-2 *Tm4sf1* and pan-reactive *Serpina3n*, *Cxcl10* and *S100a6* (Fig. 4e).

To determine if earlier treatment with PMX205 would also mediate downregulation of reactive microglia and astrocyte genes, hippocampal mRNA from a younger (7 mo) cohort of Arc mice that was treated for 2–12 weeks prior to perfusion was assessed by qPCR (Supplementary Fig. 6). While starting PMX205 treatment at 4 mo of age for 12 weeks did not significantly affect hippocampal levels of microglial marker *Cst7* or *Itgax*, reduced levels of *Tyrobp* ($p < 0.05$), *Inpp5d* ($p < 0.001$), *Lcn2* ($p < 0.001$), *Tnf* ($p < 0.05$) and *Ccl4* ($p < 0.01$) were detected compared to non-treated Arc mice. Delaying the initiation of treatment and for a shorter period of time (6 weeks) did not result in a significant reduction of inflammatory gene expression, with the exception of the astrocyte gene *Lcn2*, although 2 weeks of treatment show rapid reduction of *Tyrobp, Lcn2* and *Ccl4* ($p < 0.05$, 0.01 and 0.056 respectively). While reduced expression of microglial genes with PMX205 treatment was generally consistent with single cell RNA-seq data, expression of reactive astrocyte marker *Lcn2* was decreased in the younger cohort treated from 4.5 to 7 months, but not in the Arc mice treated with PMX205 from 7.5 to 10 months. These data suggest that while earlier treatment may be beneficial, excessive glial activation (reflected here by *Tyrobp* and *Ccl4*, microglial and astrocyte dominant, respectively) can be mitigated even in later stages of the disease by C5aR1 inhibition.

## PMX205 treatment promotes neurotrophic and neuroprotective signaling pathways in Arctic mice

Given that C5aR1 inhibition has selective suppressive effects on glial gene expression in the hippocampus of Arctic mice, CellChat was used to infer the probability and strength of intercellular communications between different brain cell populations that were affected by inhibition of C5aR1 (Supplementary Fig. 7a–d). OPCs were the greatest contributor to cell-cell signaling regardless of genotype or treatment (Supplementary Fig. 7e–h), highlighting their role monitoring and responding to their environment[40]. The number of intercellular signals and the strength of those signals was greater in the Arctic mouse relative to the wildtype, and both the number and strength of interactions was reduced by inhibition of C5aR1 in the Arctic mouse (Supplementary Fig. 7i, j).

To further interrogate altered intercellular signaling, CellChat was used to plot predicted relative information flow of significantly altered pathways in Arc vs WT and in Arc vs Arc-PMX205 cells (Fig. 5). Of the pathways that were significantly altered between WT and Arc cells (Fig. 5a), several were unique to the Arctic hippocampus and not found in WT cells, including Tenascin, ANGPT, FGF, MPZ, PROS, CDH5, and AGRN (Fig. 5b). The greater number of pathways observed in Arc compared to WT cells is reflected by higher relative number of interactions (Supplementary Fig. 7i). Other pathways relatively over-represented in Arctic compared to WT include CLDN, FN1, VTN, SEMA4, Collagen, CDH, EGF, Laminin, MAG, and PTN. Conversely, the number of pathways that were suppressed in Arc cells compared to WT were very few (CSF, PECAM1, and NRG). Of the pathways that were unique to Arc cells, Tenascin, ANGPT, PROS, and AGRN were suppressed by C5aR1 antagonism, while FGF, MPZ, and CDH5 were further enhanced by treatment (Fig. 5b and Supplementary Fig. 7j). We identified three additional pathways that were unique to Arc-PMX205 cells, which were SEMA7, TGFß, and BMP.

Interestingly, the majority of signaling that was enhanced in Arctic cells compared to WT cells derived from astrocytes (Fig. 5c). These included ANGPT, FGF, PROS, and AGRN (Fig. 5c, orange boxes). PROS signaling was also observed from microglia (Fig. 5c). However, PMX205 treatment enhanced some signaling pathways derived from astrocytes (Fig. 5c, blue boxes), including BMP and TGFß, as well as from OPCs and endothelial cells, all demonstrating the broad downstream effects of inhibition of C5aR1. Many pathways were preserved between Arc-Veh and Arc-PMX205 cells. These pathways play important roles in cell adhesion and are known to bind to the extracellular matrix (NCAM, NRXN, NRG, CADM, NEGR, CNTN, and PTN) to allow for synaptic stabilization and organization (NRXN, NRG, and PTN) (Supplementary Fig. 8). The specific ligand-receptor interactions between different cell types were also defined to determine the senders and receivers of specific signaling pathways influenced by treatment of Arc mice with PMX205 (Supplementary Fig. 9). For example, we identified enhanced FGF1 signaling from Arc-PMX astrocytes to neurons, microglia, oligodendrocytes, OPCs, endothelial cells, pericytes, and other astrocytes (Supplementary Fig. 9).

## PMX205 preserves beneficial glial responses to injury

Immunohistochemistry was used to assess plaque, microglia, and astrocyte markers in the hippocampus of the PMX205-treated vs untreated WT and Arctic mice. No differences in the pan-microglial marker Iba1 or in CD11b and CD11c, which make up part of the integrin receptors CR3 and CR4, respectively, were detected in the hippocampus between Arc-Veh and Arc-PMX (Fig. 6a–e). Additionally, no change in C5aR1 reactivity was noted with PMX205 (Supplementary Fig. 10) similar to that reported in other models[18]. Thus, combined with gene expression data above, C5aR1 inhibition in the Arc mouse selectively suppresses upregulation of some disease-associated microglial genes while having little effect on an increase of others, at least some of which may have beneficial roles (*Itgax*[41]).

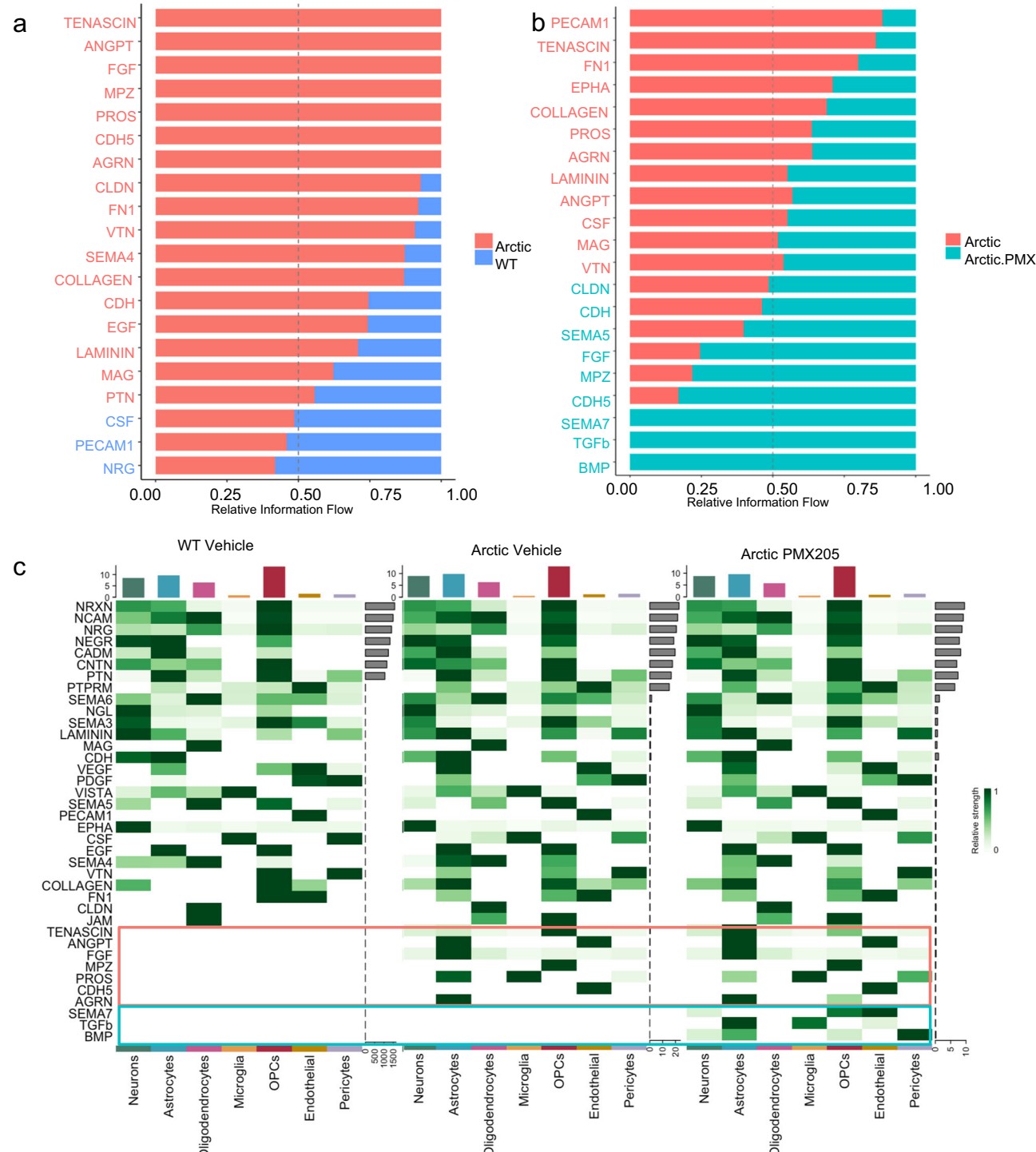

**Fig. 5 | Relative information flow of pathways significantly enhanced or suppressed in Arctic mice and altered by PMX205 treatment. a** Relative information flow of pathways that are differentially expressed between WT-veh (blue) and Arctic-veh (salmon). **b** Relative information flow of differential pathways in Arc-veh (salmon) and Arc-PMX (teal). **c** Cellular senders of differential pathways in WT-veh (left), Arc-veh (middle), and Arc-PMX (right).

PMX205 treatment had no effect on hippocampal GFAP levels (Fig. 6f–i, Supplementary Fig. 11). C3 staining showed a trend for an increase upon PMX205 treatment correlating the RNA-seq data (Supplementary Fig. 11). There was variability in the S100a6 and LCN2 field area % in the hippocampus, but no statistically significant difference between Arc and Arc-PMX (Fig. 4f–h), demonstrating a narrow, targeted effect on astrocyte polarization. Finally, mice that were treated from 7.5 to 10 months did not have a reduction in fibrillar plaque staining (Supplementary Figs. 11 and 12), indicating that inhibition of

C5aR1 signaling, while reducing detrimental gene expression, does not reduce plaque load in this aggressive model, demonstrating that it is the glial response to the plaque that dominates progression to impairment.

## Inhibition of C5a-C5aR1 signaling protects short-term spatial memory in female Arctic mice

To determine if the altered hippocampal signaling induced by C5aR1 inhibition is associated with hippocampal-dependent memory in Arc

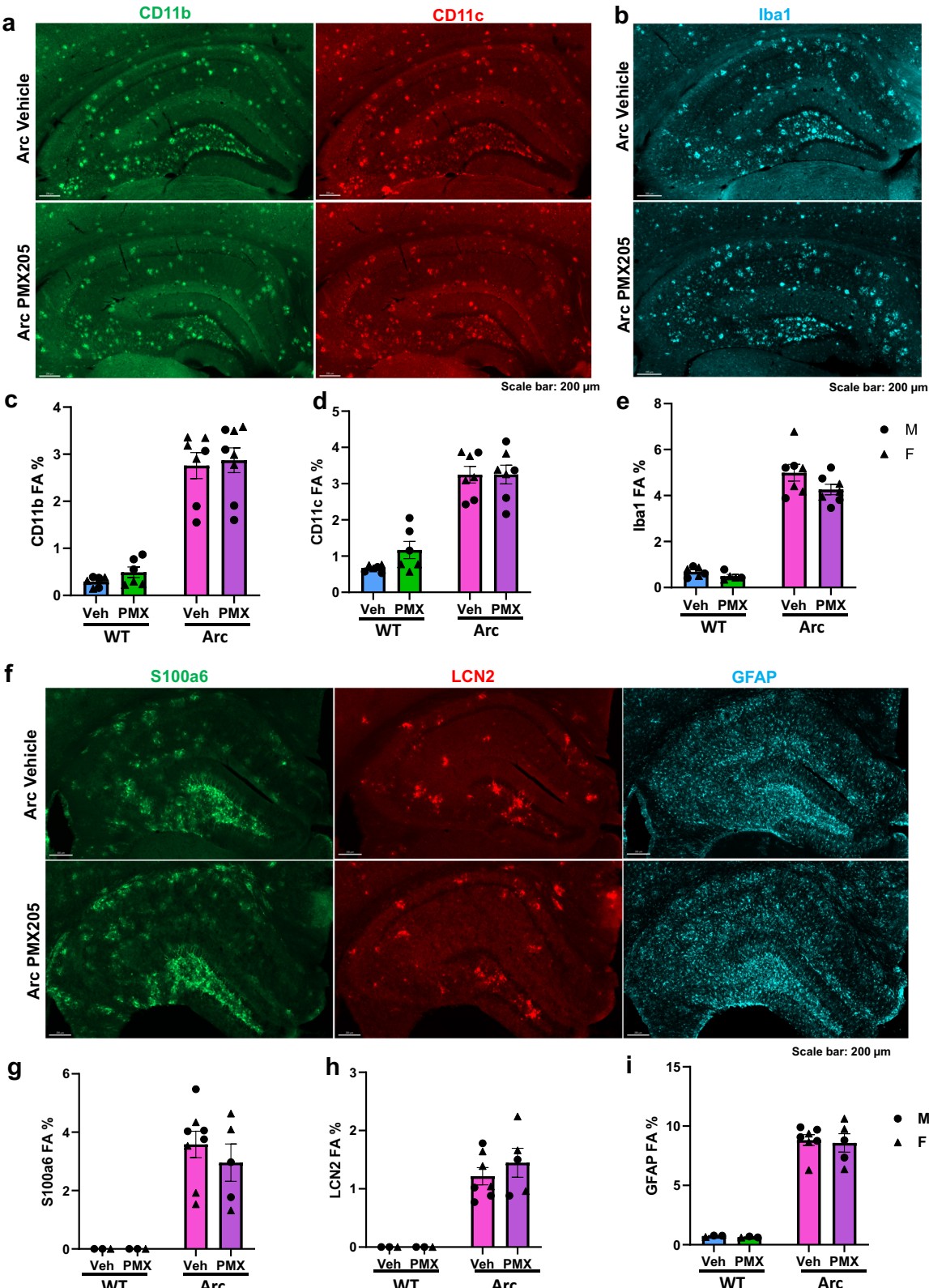

mice, we used the Y maze spatial reference test (Fig. 7) to test male and female mice. Female Arc mice spent significantly less time exploring the novel, previously blocked arm of the Y maze compared to WT mice (19% vs 29%, $p = 0.025$). However, this deficit was eliminated in female Arc mice treated with PMX205, who spent 47% of the trial in the novel arm and significantly outperformed Arc-Veh mice ($p = 0.0014$) (Fig. 7b). Male Arc mice did not show a deficit in spatial memory in the

Y maze, with most male mice exploring the novel arm 40-44% of the time allotted (Fig. 7c), suggesting that in this cohort disease progression and the onset of memory deficits is slower in male Arctic mice. The deficit in hippocampal-dependent memory in female Arc mice (prevented by C5aR1 antagonism) compared to male mice is consistent with the overexpression of DAM1 genes in microglia derived from female Arc mice compared to male Arc mice (Supplementary Fig. 3).

**Fig. 6 | PMX205 does not prevent all glial responses to injury.** Representative images of dorsal hippocampus stained for CD11b (A, left)and CD11c (**a** right) and Iba1 (**b**) in Arc-veh (top panel) and Arc-PMX (bottom panel) 10X magnification, scale bar 200 μm. **c**–**e** Quantification of percent field area (FA %) of WT-veh, WT-PMX, Arc-veh, and Arc-PMX for CD11b (**c**), CD11c (**d**), and Iba1 (**e**). **f** Representative images of dorsal hippocampus stained for S100a6 (left), LCN2 (middle), and GFAP (right) in Arc-veh (top panel) and Arc-PMX (bottom panel) 10X magnification, scale bar 200 μm. Quantification of FA% of WT-veh, Arc-veh, and Arc-PMX for S100a6 (**g**), LCN2

(**h**), and GFAP (**i**). Males (circles) and females (triangles) are designated in all graphs. Data shown as mean ± SEM of 3-4 images per mouse. Two-way ANOVA, $n$ = 3-8 mice/ genotype/treatment (CD11b, $n$ = 6 WT-veh, $n$ = 6 WT-PMX, $n$ = 7 Arc-veh, $n$ = 8 Arc-PMX. CD11c, $n$ = 6 WT-veh, $n$ = 6 WT-PMX, $n$ = 7 Arc-veh, $n$ = 7 Arc-PMX. Iba1, $n$ = 6 WT-veh, $n$ = 5 WT-PMX, $n$ = 7 Arc-veh, $n$ = 7 Arc-PMX. S100a6 $n$ = 3 WT-veh, $n$ = 3 WT-PMX, $n$ = 8 Arc-veh, $n$ = 5 Arc-PMX. LCN2 $n$ = 3 WT-veh, $n$ = 3 WT-PMX, $n$ = 7 Arc-veh, $n$ = 5 Arc-PMX. GFAP $n$ = 3 WT-veh, $n$ = 3 WT-PMX, $n$ = 7 Arc-veh, $n$ = 5 Arc-PMX.).

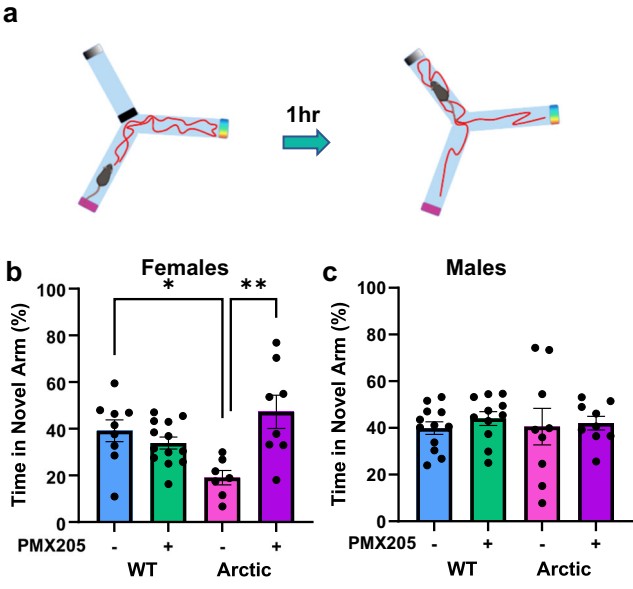

**Fig. 7 | PMX205 protects against spatial memory deficits in Arctic females.** **a** Overview of Y maze experiment completed at 10 months of age. Time spent in the novel arm during the test trial in females (**b**) and males (**c**). Data shown as mean ± SEM. *$p$ < 0.05; **$p$ < 0.01, Two-way ANOVA with Tukey's *post hoc*. WT-veh vs Arc-veh, $p$ = 0.0253; Arc-veh vs Arc-PMX, $p$ = 0.0014. $N$ = 12M9F (WT-veh), 11M13F (WT-PMX), 9M7F (Arc-veh) and 9M8F (Arc-PMX).

Finally, plasma NfL levels were measured before and after PMX205 treatment as a biomarker for axonal injury. Surprisingly, PMX205 treatment did not alter plasma NfL levels in Arc mice compared to untreated mice (Fig. 8a). To determine if earlier onset of treatment would yield a stronger effect, we assessed plasma NfL in our smaller cohort of younger Arc mice treated for 2, 6, or 12 weeks (4.5–6.5 months to 7 months of age) with PMX205. In this younger cohort, Arc- H₂O mice had higher levels of NfL compared to WT mice at 7 months of age (p < 0.0001) and compared to their pretreatment levels (at 4.5 months). Treatment with PMX205 for 2- (*$p$ < 0.05*), 6- (*$p$ < 0.01*), or 12- (*$p$ = 0.13*) weeks prevented this time-dependent increase (Fig. 8b). Thus, C5aR1 antagonism may be most beneficial when started at early stages of plaque accumulation.

## Discussion

Complement activation contributes to proper synaptic connectivity and plasticity during development as well as in adults[42–44]. However, with age and injury, complement pathway components are induced in the brain and when activated can lead to excessive synaptic pruning, neuroinflammation and cell death[11,45]. Activation of the classical complement pathway can be triggered by interaction of C1q in the C1 complex by fibrillar Aβ plaques, neurofibrillary tangles as well as the exposure of damage-associated molecular patterns on neuronal

cell surfaces[46–49], all of which can lead to downstream production of C5a, C3a and C5b-9. In brain, C5aR1 and C3aR are prominently induced in microglia in response to injury (reviewed in ref. 50). If complement is activated and C5 is present locally to enable production of C5a, C5aR1 signaling can initiate pro-inflammatory patterns of gene expression. Here, we demonstrate that pharmacologic inhibition of C5a-C5aR1 signaling even after substantial plaque accumulation improved cognitive outcomes in the aggressive Arctic mouse model of AD, decreased transcription of disease-enhancing microglial genes while retaining protective gene expression and altered astrocyte polarization.

Inhibition of C5aR1 signaling has been shown to suppress several parameters of glial activation in the slower progressing Tg2576[21,22] and 3xTg[22] AD mouse models. However, in those models C5aR1 inhibition decreased amyloid plaque accumulation by 50%. Here, in the more aggressive Arctic model, plaque load does not change upon inhibition of C5aR1, allowing a distinction between C5a-C5aR1-induced effects on the cellular response to amyloid pathology versus the plaque-induced effects that are independent of C5aR1 (Supplementary Fig. 13). In addition, in a more translationally relevant pharmacologic inhibition of C5a-C5aR1 signaling, versus previous studies with genetic ablation of C5aR1[20,51], cognitive outcomes improved even when treatment was started well after the accumulation of amyloid, consistent with the hypothesis that inhibition of C5aR1 dampens detrimental amyloid-induced inflammatory responses. Here both single cell RNA-seq of microglia and single nuclei RNA-seq (used to obtain information about other cell types) provide insight on communication among hippocampal cellular subtypes as well as the molecular basis for the neuroprotective effects that result from inhibition of C5a-C5aR1 signaling.

Importantly, specific and previously undefined microglial and astrocyte clusters were identified as displaying consequences of the expression of the APP transgene that were C5aR1-independent and those that were dependent on C5aR1 signaling. Microglia clusters M4 and M9 were C5aR1-dependent and expressed genes that detrimentally modulated synapse pruning, microglial activation, cognition and cell death, whereas clusters M3 and M8 which had high expression of genes previously characterized as DAM-2 genes were induced in the Arctic without C5aR1 input (C5aR1-independent), and may have repair or restorative functions which are advantageous to maintain, rather than suppressed by more globally targeting strategies. In addition, M2, which contributes to organization of synapses, is suppressed in C5aR1-sufficient Arctic mice but is rescued by C5aR1 inhibition. These findings, that C5aR1 antagonism in Arc mice enhances pathways associated with synaptic plasticity and learning, importantly provide a mechanistic basis for the wide range of human and mouse studies demonstrating that some microglial responses are attempts to protect from injury and should not be inhibited (reviewed in refs. 52,53).

While differences were not as extensive in astrocytes, astrocyte clusters were differentially affected by the Arctic pathology and inhibition of C5aR1. For example, astrocyte cluster A6, a neuronal supportive cluster associated with axon guidance and synapse transmission, was suppressed in Arctic, but rescued with inhibition of C5aR1 signaling. Secondly, A3 astrocytes, expressing disease-mitigating genes, were induced in response to the APP transgene regardless of C5aR1 function (C5aR1-independent and therefore not negatively affected by PMX205). Finally, A7, disease-enhancing astrocytes, was induced in the Arctic

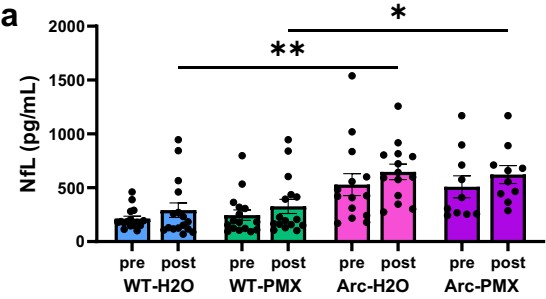

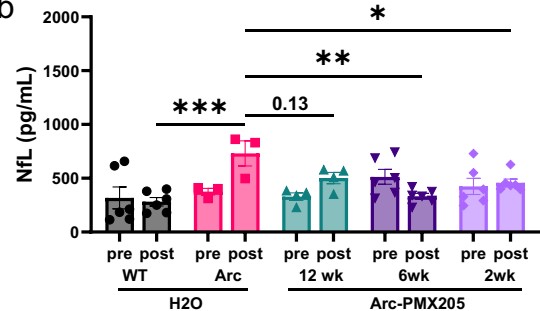

**Fig. 8 | Early treatment with PMX205 suppresses neurofilament light levels in plasma of Arc mice. a** Plasma NfL levels of WT and Arctic mice before (pre) and after (post) PMX205 or water treatment from 7.5 to 10 months. **b** Plasma NfL levels in WT and Arc pretreatment and at 7 mo after (post) PMX205 treatment for 2, 6, or 12 weeks. Data shown as mean ± SEM. *$p < 0.05$; **$p < 0.01$; ***$p < 0.001$, Repeated measures Two-way ANOVA with Tukey's post hoc, WT-H$_2$O (post tx) vs Arc- H$_2$O

(post tx) $p = 0.002$; WT-PMX (post tx) vs Arc-PMX (post tx) $p = 0.0316$. N = 16 (WT-H$_2$O), 16 (WT-PMX), 14 (Arc- H$_2$O) and 10 (Arc-PMX) (**a**) and Sidak's post hoc, Arc vs WT, $p = 0.0004$; Arc vs Arc-PMX 12 wk $p = 0.13$; Arc vs Arc-PMX 6 wk $p = 0.0018$; Arc vs Arc-PMX 2 wk $p = 0.0486$. N = 6 (WT- H$_2$O), 3 (Arc- H$_2$O), 4 (Arc-PMX 12 wk), 6 (Arc-PMX 6 wk), and 6 (Arc-PMX 2 wk) (**b**). Males and females were included in both studies with no apparent sex differences evident in any genotype/treatment group.

mouse but completely absent in Arctic treated with PMX205, i.e. was C5aR1 dependent. Disease-specific changes in astrocyte gene expression have also been reported in snRNA-seq data from human AD samples[54]. Astrocyte activation has long been proposed to have positive effects in the CNS response to injury[55]. The data here are consistent with a model in which C5aR1-induced signals generated by microglia influence the activities of nearby astrocytes. Interestingly, *C3* expression, previously labeled as an "A-1" disease-associated astrocytic protein[34], while increased as expected in the Arctic mouse, is further increased by inhibition of C5aR1. While often considered as enhancing inflammation and excessive synapse pruning due to extracellular complement activation, a role for intracellular C3 as a metabolic regulator for cells of the immune system has been documented[56]. It has been hypothesized that glial-derived C3 may be involved in intracellular signaling and/or metabolic reprogramming[57] such that induction of astrocyte C3 under stress conditions (and thus a "disease associated" astrocyte protein) may very well have beneficial/mitigating effects, particularly if it enhances metabolism and thus supports astrocytic neuroprotective roles.

Although not the primary objective of this study, the single nucleus RNA-seq clearly demonstrated upregulation of individual complement proteins in specific clusters of cell type subsets that are induced by pathology. *C1q* was predominantly induced in M4 microglia, *C3* in A3 and A10 astrocytes, and *C4b* (the mouse *C4* gene) in both astrocytes and oligodendrocytes, supporting earlier reports of these individual components[34,58–61]. C5 (*Hc*) was detected in our samples at very low levels and in a small number of neurons and astrocytes, as reported by others[16,61–63]. In contrast, almost all clusters of neurons expressed the complement control proteins *Csmd1* and *Csmd2* (and captured OPCs), suggesting a cell autonomous regulation of complement activation by neurons. Both *Csmd1* and *Csmd2* variants have been associated with multiple disorders including those associated with cognition[64–66].

While *C5ar1* expression is dominant in activated microglia and peripheral myeloid cells, CellChat analysis of hippocampal snRNA-Seq data revealed that inhibition of C5a-C5aR1 signaling affected several intercellular pathways associated with not only microglia, but also astrocytes and oligodendrocytes, consistent with a complex cellular phase of AD initiated by C5aR1-induced secreted mediators[67], that may be blunted by this targeted approach in multiple subtypes of AD[4,5]. CellChat revealed that PMX205 treatment in Arc mice promoted signaling of fibroblast growth factors (FGFs) from astrocytes. FGFs comprise a large family of polypeptides that are essential for cell development, maintenance, and repair[68]. FGFs promote astrocyte differentiation and proliferation[69]. In models of spinal cord injury, FGF2 is secreted by astrocytes to signal back to other astrocytes for regulating astrogliosis and promoting structural changes in glia, and

promotes survival of OPCs[69]. FGF2 has been shown to be down-regulated in the prefrontal cortex of AD patients[70]. In the APP23 mouse model of AD, overexpression of FGF2 is neuroprotective, restoring hippocampal-dependent spatial memory and reducing levels of Aβ and the APP cleaving enzyme (BACE)[70]. Attenuation of cognitive decline by FGF may involve stimulation of hippocampal neurogenesis[71]. However, in vitro studies show that glutamate- or oligomeric Aβ-induced toxicity promotes FGF2 secretion by neurons, and this event promotes microgliosis and phagocytosis of neuronal debris[72].

TGFβ2 signaling from astrocytes to microglia and endothelial cells was also enhanced in PMX205-treated Arctic mice. TGFβ is a super family of proteins that includes TGFβ, BMPs, and activins. TGFβ binding to its receptor activates SMAD3, which promotes amyloid-β uptake and iNOS production by microglia and astrocytes, respectively[73]. TGFβ signaling is important for microglial development and promotes expression of quiescent microglial genes including *P2ry12*, *Fcrls*, and *Sall1*[73]. Furthermore, microglia act via TGFβ to promote hippocampal neurogenesis in inflammatory or neurodegenerative contexts[74]. Cell-cell communication from pericytes to astrocytes was increased in Arc mice with PMX205 treatment relative to vehicle-treated Arc mice as is the BMP pathway, supporting the possibility that with C5aR1 inhibition, pericytes send trophic or survival factors, such as BMP5, to astrocytes. Based on snRNA-Seq, CellChat predicts BMP5 signaling to be present in pericytes isolated from Arc-PMX hippocampi, with signaling directed towards astrocytes, neurons, and OPCs. Notably, this pathway was absent in Arc-Veh cells. Finally, evidence from preclinical models of ischemic stroke suggest that reactive astrocytes upregulate expression of tenascin to inhibit proliferation, possibly as a self-limiting mechanism[75]. The shift of tenascin signaling from OPCs to astrocytes in the Arc-PMX205 cells, in addition to FGF signaling, suggests that inhibition of C5aR1 promotes a return to homeostatic state in astrocytes exposed to amyloid pathology.

The complete genetic knock out of C5aR1 results in protection from loss of spatial memory[51] and rescues region specific pre-synaptic loss[45] in AD models. The results presented here show that pharmacologic inhibition of C5aR1 in adult mice with substantial amyloid plaque load can also protect against hippocampal-dependent spatial memory deficits, highlighting its potential as a therapeutic target to slow or ameliorate cognitive decline in adults. While suppression of plasma NfL, a biomarker of neuronal injury, was noted at 7 months of age after 2-12 weeks of PMX205 treatment, unexpectedly, plasma NfL at 10 months of age was not impacted by C5aR1 inhibition. Thus, C5aR1 inhibition via PMX205 may be most effective if initiated at earlier stages of the disease for maximum effect, consistent with changes in human AD microglial states with disease progression[7].

C5aR1 has become a target of interest for several neuroinflammatory conditions[76–78]. Clinical and preclinical studies reported increased complement proteins in the brain and CSF in amyotrophic lateral sclerosis, stroke, epilepsy, traumatic brain injury, multiple sclerosis, and Alzheimer's disease (reviewed in refs. 8,12). Importantly, evidence supports that the less understood C5a receptor 2 (C5aR2) prevents lesion formation and promotes recovery after spinal cord injury[31]. Thus, targeting the pro-inflammatory receptor C5aR1 would be more beneficial than targeting the ligand C5a, which may interact with C5aR2 to promote healing. It is critical to note that complement is an important protective system against infection, and in clearing dead cells and debris[79], and thus complete long-term inhibition of the system especially C1q through C3, while also blocking the downstream generation of C5a, would have significant immunocompromising effects. The 2021 FDA approval of the small molecule C5aR1 antagonist, avacopan (Tavneos) for treatment of the peripheral disorder, antineutrophil cytoplasmic antibody-associated vasculitis, as well as earlier small studies of PMX53 in humans, suggests a lack of toxicity when blocking this receptor in humans although with the possible reduced response to systemic candidemia[27,80]. Whether the newly approved avacopan or PMX205 will be effective in substantially slowing the progression of cognitive decline in Alzheimer's disease or other neurodegenerative disorders awaits human clinical trials.

In summary, treatment with PMX205 when plaque accumulation is robust was sufficient to attenuate the inflammatory response to the plaques and preserve short-term spatial memory. Unique clusters of microglia and astrocytes were identified as being C5aR1-dependent, while other beneficial clusters induced in the present of amyloid plaques were C5aR1-independent and thus not affected by treatment with the antagonist. Assessment of RNA expressed by other brain cell types supports extensive communication between cell types that is also modulated by C5aR1 in the context of amyloidosis. Supported by observations from several animal models and positive effects in humans, it can be concluded that inhibition of C5aR1 reduces inflammatory responses while preserving protective functions of complement activation. Thus, targeting C5aR1 has high potential for treatment of neurodegenerative diseases, and therefore is a promising candidate for clinical trials in disorders such as AD.

## Methods

### Animals
The Institutional Animal Care and Use Committee of University of California at Irvine granted ethical approval for all animal procedures (#AUP-21-132). Experiments were performed according to the NIH Guide for the Care and Use of laboratory animals. Mice were single housed in ambient temperature and given access to food and water *ad libitum*. The Arctic48 mouse model of AD (hereafter referred to as Arc), which carries the human APP transgene with three mutations – the Indiana (V717F), the Swedish (K670N + M671L), and the Arctic (E22G), was generated on a C57BL6/J background and originally provided by Dr. Lennart Mucke (Gladstone Institute, San Francisco, CA). Mice were crossed with C57BL6/J wild-type (WT) mice to create Arc+/− and Arc−/− (WT) mice. The Arctic48 APP mutation results in rapid fibrillar formation, a conformation that is more stable and/or difficult to clear. As a result, this hemizygous mouse model produces fibrillar plaques as early as 2–4 months of age[20,81].

### PMX205 treatment
C5aR1 antagonist PMX205 (Mimotopes, Victoria, Australia) was diluted in MilliQ $H_2O$ to a final concentration of 20 μg/ml. To fully solubilize PMX205 from powder form, we dissolved 5 mg in 1 mL of MilliQ $H_2O$, then diluted to the final concentration. Drinking bottles were filled with 100 mL of PMX205 or vehicle (MilliQ $H_2O$) and were weighed and refilled weekly. Mice were single housed throughout treatment in order to calculate drug consumption for each mouse. Mice were also weighed weekly to detect potential toxic effects of PMX205 treatment.

In a smaller pilot experiment, Arc mice were treated with PMX205 via drinking water for 2, 6, or 12 weeks beginning at 6.5, 5.5 or 4.5 months of age, respectively to detect biomarker changes. A cohort of age-matched WT and Arc mice were untreated and used as controls. For all cohorts, plasma samples were collected prior to treatment by submandibular puncture under anesthesia and at perfusion via cardiac puncture. Brains were perfused and collected for IHC, and hippocampi were dissected for RNA-seq and qPCR assays.

### Y maze spatial reference test
Mice were subjected to behavioral testing following PMX205 or vehicle treatment at 10 months of age. Female mice were tested before male mice, and equipment was thoroughly cleaned or changed between sex groups to eliminate odor cues. The Y maze test for spatial memory reference was adapted from ref. 82 to measure short-term hippocampal-dependent spatial memory. Briefly, mice were placed into an arm of a three-armed maze, with one of the arms blocked off and given 5 min to explore, after which mice were placed back in their home cages. After a 1-h intertrial interval, the block from the third arm was removed, and mice were again given 5 min to explore freely. Time spent in each arm was tracked with Noldus EthoVision. We compared total time spent in the novel arm between groups and preference for the novel arm over the two familiar arms within groups with Two-way ANOVA and Tukey's post hoc test. This test took place in normal (200 Lux) lighting. The ends of each arm had distinct visual patterns so that mice could distinguish between a novel and familiar arm. The mazes were cleaned with 70% ethanol and dried thoroughly after each trial.

### Neurofilament light assay
Blood was collected before treatment via submandibular puncture and before perfusion via cardiac puncture for plasma isolation. Blood was immediately mixed with 0.2 M EDTA to a final concentration of 10 mM and centrifuged at 2400 × $g$ (Eppendorf Centrifuge) for 10 minutes at 4 °C. Supernatant (plasma) was collected and stored in −80 °C until use. To measure plasma neurofilament light (NfL) levels, the MesoScale Diagnostics NfL kit (F217X-3) was used following manufacturer's instructions. Samples were assessed in duplicate.

### Microglia isolation and fixation
Mice were deeply anesthetized with isoflurane, perfused transcardially with HBSS (without $Ca^{2+}$, $Mg^{2+}$) and half brains dissected to separate cortex and hippocampi[21]. Half hippocampi from 2–3 mice of the same sex, genotype and treatment were pooled per sample for processing in HBSS with an inhibitor cocktail to prevent artificial microglial activation (actinomycin D 5 μg/ml, anisomycin 27.1 μg/ml, and triptolide 10 μM)[83] (3 samples per sex/genotype/treatment). To isolate microglia from hippocampi, we followed manufacturer protocols from Miltenyi Biotech[21,83]. Briefly, cells were incubated with anti-CD11b beads, and labeled microglia were collected in LS columns and then eluted from the columns. Microglia were centrifuged at 500 × $g$ for 10 min at 4 °C, resuspended in 375 μL of cell buffer from the kit with 0.5% BSA, and counted before continuing to the cell fixation protocol provided by Parse Biosciences (V1.3.0). After incubation in cell permeabilization solution, 2 mL of cell neutralization buffer were added to each sample and they were centrifuged at 500 × $g$ for 10 minutes at 4 °C. Pellets were resuspended in cell buffer with DMSO (1:20 dilution), counted again, aliquoted, and stored at −80 °C until ready for barcoding.

### Nucleus isolation and fixation
Nucleus isolation was done with Nuclei Extraction Buffer (Miltenyi Biotech 130-128-024) following manufacturer's instructions. Briefly,

perfused half hippocampi (3 per sex/genotype/treatment) were collected and frozen at −80 °C until use. Frozen hippocampi were placed directly into C-tubes with 2 mL lysis buffer (Nuclei Extraction Buffer with 0.2 U/µL RNase inhibitor) and processed in the Miltenyi gentleMACS™ Octo Dissociator. Nuclei were passed through a 70 µm SmartStrainer. An additional 2 mL of lysis buffer was added to the C-tubes and passed through the strainer to capture more nuclei. Samples were then centrifuged at $350 \times g$ at 4 °C for 5 min. Nuclei were resuspended in 1.5 mL of resuspension buffer (PBS with 0.1% BSA and 0.2 U/µL RNase inhibitor), passed through a 30 µm filter, and counted before proceeding to nuclei fixation protocol (Parse Biosciences V1.3.0). 4 million nuclei were collected from the stock of each sample preparation, centrifuged, and resuspended in 750 µL nuclei buffer with 0.75% BSA (Parse Biosciences). Samples were passed through a 40 µm filter, incubated in nuclei fixation solution for 10 min on ice, followed by incubation in nuclei permeabilization solution. 4 mL of nuclei neutralization buffer were added to each sample, and they were centrifuged at $500 \times g$ for 10 min at 4 °C. Pellets were resuspended in nuclei buffer with DMSO (1:20), frozen and stored at −80 °C.

## Barcoding and library preparation for single cell and single nuclei RNA-seq

The Evercode Whole Transcriptome (WT) kit (Parse Biosciences V1.3.0) was used to prepare single nucleus and single cell microglia libraries for RNA-seq. Forty-eight samples (24 single cell and 24 single nucleus) diluted to 525 cells or nuclei/µL were each loaded onto a separate well in a 96-well plate provided by Parse Biosciences for initial barcoding. Following reverse transcription and barcoding, all SC and SN samples were pooled and incubated in ligation mix for a second round of barcoding. The pooled cells were then redistributed to a 96-well plate. This step was repeated once more for a total of 3 barcoding cycles. The cell/nuclei mix was then pooled again and counted to calculate final concentration. After cell counting, 7 separate aliquots of 12,500 cells or nuclei per sublibrary were processed in parallel, for a combined 87,500 cells/nuclei total. For each sublibrary, cells/nuclei were lysed, barcoded cDNA was amplified, fragmented, and a unique sublibrary index was added prior to sequencing. The seven sublibraries were sequenced on a single NextSeq2000 (Illumina) sequencing run for an average of 10902reads/cell. Resulting Fastq files and data matrices were deposited in GEO with the accession ID GSE240950.

## RNA-seq processing and data analysis

Fastq files were processed using the Parse short-read split-pipe pipeline[84]. Doublets were removed using Scrublet[85]. Cells and nuclei were filtered using different metrics from the matrices generated. Cells were filtered for more than 300 UMIs, <10% mitochondrial reads, <50,000 counts and >300 genes. Nuclei were filtered for more than 300 UMIs, <1% mitochondrial reads, <50,000 counts and > 400 genes. Then, the cells and nuclei were integrated with Seurat. In downstream analyses, normalization and clustering were calculated using Seurat SCTransform and Leiden algorithms, respectively[86]. Wilcoxon Rank Sum test was performed to identify differentially expressed genes between two or more groups of cells. *P* value adjustment was performed using Bonferroni correction based on the total number of genes in the dataset. Gene ontology (GO) analysis was performed for each Seurat cluster using Metascape[87] by inputting cluster genes with adjusted *p* value <0.05. Intercellular communication between clusters of cells was inferred using CellChat[88]. Sequencing data is freely available in GEO with the accession ID GSE240950. Codes are available on GitHub (https://github.com/heidiiliang/R-code-for-ArcPMX205/tree/main).

## Quantitative PCR

Frozen hippocampi from 7 mo WT ($n = 6$) and Arc H$_2$O controls ($n = 4$), and Arc mice treated for 2 ($n = 6$), 6 ($n = 6$), and 12 ($n = 5$) weeks with

PMX205 were pulverized using mortar and pestle for RNA quantification of target genes[17]. The following FAM dye and MGB quencher TaqMan probes were used (ThermoFisher): Mm00438349_m1 (*Cst7*), Mm00498708_g1 (*Itgax*), Mm01324470_m1 (*Lcn2*), Mm00443260_g1 (*Tnf*), Mm0043111_m1 (*Ccl4*), Mm00449152_m1 (*Tyrobp*), and Mm00494987_m1 (*InppSd*). All probes were multiplexed with VIC dye and MGB quencher probe Mm01545399_m1 (*Hprt*) as an internal control for each well. cDNA from each hippocampus was tested in triplicate. Within each well, relative expression to *Hprt* was calculated by subtracting the VIC cycle threshold (Ct) value from FAM Ct value. ΔCt values within triplicates were averaged then exponentially transformed and multiplied by 1000 ($2^{-\Delta Ct} * 1000$). *P* values were calculated using one-way ANOVA followed by Dunnet's multiple comparisons test.

## Immunohistochemistry

Mice were deeply anesthetized with isoflurane, perfused transcardially and half brains dissected, fixed in 4% paraformaldehyde for 24 hr, stored in PBS 0.1% sodium azide and 30 µm coronal sections were obtained for immunohistochemistry. After incubating sections in blocking solution (2% BSA, 10% normal goat or donkey serum, 0.1% Triton PBS) tissues were incubated in the following primary antibodies overnight at 4 °C with agitation: rabbit anti-Iba1 (1:1000, Wako #019-19741), rat anti-CD11b (1:1000, Biorad #MCA74G), hamster anti-CD11c (1:400, Biorad #MCA1369), goat anti-Lcn2 (1:50, R&D #AF1857), chicken anti-GFAP (1:1000, Abcam #ab4674), and rabbit anti-S100a6 (1:500, Novus #NBP1-89388), rat anti-CD88 10/92 (1:1000, BioRad #MCA2456), rat anti-C3 clone 11H9 (1:50, Hycult #HM1045), and rabbit anti-GFAP (1:2900, Dako #Z0334). Following washes in 0.1% Triton PBS, tissues were incubated in the appropriate Alexa Fluor secondary antibodies, diluted 1:500 in blocking solution[20]: goat anti-Armenian hamster alexa 568 (Abcam, # ab175716), goat anti-rabbit alexa 488 (Invitrogen #A11070), goat anti-rat alexa 488 (Invitrogen #A-11006), goat anti-rabbit alexa 555 (Invitrogen #A-21429), goat anti-rat alexa 555 (Invitrogen #A21434), goat anti-rabbit alexa 647 (Invitrogen #A-21244), donkey anti-goat alexa 555 (Invitrogen #A-21432), donkey anti-rabbit alexa 488 (Invitrogen #A-21206), and donkey anti-chicken alexa 647 (Jackson Immuno Research #803605155). Antigen retrieval buffer (50 mM TBS, 0.05% Tween, pH 9) was used before staining for the C5aR1 protein. To visualize plaques tissues were incubated for 10 min with Amylo-Glo (1:100 in PBS, Biosensis #TR-300-AG) or Thioflavine S (0.5% in milliQ water, Sigma, #T1892) for 10 min. Low magnification images (10X) were acquired using ZEISS Axio Scan.Z1 Digital Slide Scanner. The field area percent of markers for microglia, astrocytes, and plaques were quantified using the Surfaces feature of Imarisx64 (version 9.5.0). Quantitative comparisons between groups were carried out on multiple sections of each animal and processed at the same time with the same batches of solutions.

## Statistics and reproducibility

Power analysis was performed a priori to determine sample size needed to achieve power of 0.8 for behavioral assays using G*Power. Mice were randomly assigned to experimental groups and experimenters were blinded to allocation during testing and analysis. Statistical analyses were performed with GraphPad Prism (V 9.5.0). Treatment X genotype effects were analyzed using two-way ANOVA with Tukey's *post hoc* test. Pre- and post-treatment effects were compared with repeated measures ANOVA with Sidak's *post hoc* test. Data that were greater than 2 SD from the mean were excluded as outliers and are included in the source data file.

## Reporting summary

Further information on research design is available in the Nature Portfolio Reporting Summary linked to this article.

## Data availability

All transcriptomic data are available in the supplementary materials. Fastq files and data matrices were deposited in GEO with the accession ID GSE240950. Source data are provided with this paper.

## Code availability

Codes are available on GitHub (https://github.com/heidiiliang/R-code-for-ArcPMX205/tree/main).

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

## Acknowledgements

This study was supported by National Institutes of Health grant R01 AG060148 (AJT and AM), Alzheimer's Association Research Fellowship AARFD-20-677771 (N.D.S.), Larry L. Hillblom postdoctoral fellowship #2021-A-020-FEL (A.G.A.), T32 AG00096 (T.J.P.), and the Edythe M. Laudati Memorial Fund (A.J.T.). This study was made possible in part through access to the Optical Biology Core Facility of the Developmental Biology Center, a shared resource supported by the Cancer Center Support Grant (CA-62203) and Center for Complex Biological Systems Support Grant (GM-076516) at the University of California, Irvine. The authors thank Dr. Whitney England for helpful comments on the data analyses.

## Author contributions

N.D.S., A.J.T., and A.M. conceived of and planned the research reported. A.J.T. and A.M. oversaw all aspects of the study. N.D.S., H.Y.L., K.C., S.C., A.M.A., T.J.P., and A.G.A. performed and analyzed the experiments. N.D.S., H.Y.L., and S.C. performed cell and nuclei isolation for sn- and scRNA-seq, which was then performed by H.Y.L. K.C., H.Y.L., and A.M. analyzed and generated figures for transcriptomics and CellChat. N.D.S., S.C., T.P., and A.G.A. performed immunohistochemistry and image analysis. A.M.A. performed qPCR and N.D.S. performed and analyzed NfL MSD. N.D.S. and A.J.T. wrote the original draft of the manuscript, with substantial input, review and edits from all authors. A.J.T., A.M., N.D.S., A.G.A., T.J.P. secured funding.

## Competing interests

The authors declare no competing interests.
