## [Peer Review File · Nature Communications]

C5aR1 antagonism suppresses inflammatory glial responses and alters cellular signaling in Alzheimer's disease mouse modelREVIEWER COMMENTS

Reviewer #1 (Remarks to the Author):

Schartz et al. report the beneficial effects of treatment with a complement C5aR1 antagonist called PMX205 in the Arctic mutant APP mouse model of Alzheimer's disease. Interestingly, the drug prevented cognitive decline in female Arctic mice without reducing amyloid, suggesting that the mechanism behind the drug's efficacy may be due to alterations in the immune response and cell signaling. To address this question, the authors performed single cell and single nucleus RNAseq on hippocampal tissues comparing Arctic mice with and without PMX205 treatment. PMX205 treatment caused robust changes in microglial genes including downregulation of disease-associated microglia and upregulation of microglial genes associated with synaptic health and learning. PMX205 also reduced expression of some reactive astrocyte genes. Overall, PMX205 promoted cell growth and repair and attenuated inflammatory pathways. The authors emphasize that this beneficial effect of PMX205 leaves the rest of the complement system intact to defend against pathogens. The methods and data/figures are clear and well-described.

Overall, this is a very well-written, highly significant paper in terms of both mechanisms of neuroprotection and support for targeting C5aR1 as a treatment for Alzheimer's disease. Importantly, this treatment strategy has implications for many other neurodegenerative diseases beyond Alzheimer's. Understanding the cellular pathways that underlie its beneficial effects may lead to additional therapeutic targets.

Comments:

1. Line 35 (Abstract): "C5aR1 inhibition promoted signaling pathways" – do you mean mRNA or protein? Please clarify here.
2. PMX205 reduced cerebral Aβ levels in Tg2576 mice but not in Arctic mice. Any idea of why there was this difference between mouse models? Do you think this was due to the timing of the intervention relative to disease stage or differences in the models?
3. Did PMX205 have any effect on CAA? This would be very interesting because CAA in human brain is associated with complement activation.
4. Line 61-62: Didn't Paul Morgan's group find C5b-9 colocalization with plaques? If so, their work should be cited here.
5. Line 100-101: Why were WT and Arc mice treated with different doses of PMX205. Please explain the rationale for this. Was it due to weight differences between the mouse lines?
6. In Figures 2 and 4, it appears that PMX205 increases C3 expression in astrocytes. Typically, C3 upregulation is associated with an A1 (toxic) astrocyte response. How do the authors reconcile this seemingly pro-inflammatory effect of PMX205 in astrocytes? Please comment on this in the Discussion.
7. Any idea of why the DAM transcript changes were mostly seen with scRNAseq compared to snRNAseq?
8. Did PMX205 have any effect on C5a levels? What about C5aR2 expression?

Reviewer #2 (Remarks to the Author):

Beneficial effects of C5aR1 antagonists have been shown before for therapeutic approaches of AD, ALS, spinal cord injury and other CNS disease, but the exact mechanisms were unknown. The authors' previous work, using a different AD model (TG2576) but the same C5aR1 antagonist (PMX205), showed a decrease in a unique microglia population responsible for synaptic pruning and increase of DAMP2 gene expressing microglia population associated with neuroprotection.

These present studies provide further data that pharmacologic inhibition of C5aR1 prevents deficit in short-term memory of female mice in an aggressive model of AD. A more detailed characterization of microglia and astrocyte gene expression shows more microglia specific changes and activation of cell growth and repair associated pathways including TGFb, bone morphogenetic protein (BMP) and FGF activation in parallel with suppression of inflammatory pathways, such as PROS, Pecam1 and EPHA. While data presented in this paper might seem to be incremental, it is conclusive in demonstrating the beneficial effect of PMX205 in advanced amyloid pathology when substantial amyloid plaques are already present, highlighting its potential utility in AD intervention therapy.

The therapeutic effect in treated animals despite the absence of any impact on amyloid plaque load is interesting and is presented by the authors as an alteration of C5aR1-mediated reactivity/immune response to pre-existing amyloid plaques. While significant steps are still needed to understand the full extent of the complement activation pathway in human AD, this pharmacological target differs from the more direct plaque targeting drugs which have been investigated extensively and could prove to work synergistically with other candidate drugs. However, significant reductions in plaque accumulation in other models (3xTg and Tg2576) are still seen (up to 50%) with PMX205 and the failure of PMX205 to influence plaque load in Arctic mice was only briefly discussed.

CellChat analysis led to the most intriguing data of this paper. According to this analysis, OPCs are the most prominent contributors to increased cell-cell signaling in AD that supports the highly adaptive and monitoring nature of these cells. It might be helpful to consider additional approaches to understand the significance of this information.

The paper is significant because this is the first study to provide a detailed gene expression profile analysis of multiple cell types during C5aR1 antagonist treatment in the aggressive arctic model of AD. However, despite multiple gene expression profiles being described, in the absence of deeper mechanistic studies, the overall mechanism and translational value are difficult to appreciate. Most of the data supports earlier reports of different mechanisms regarding neuroprotective effects.

Minor comment: Fig. 6A is missing "c" from CD11

Fig. 5: SEMA7 should be SEMA7A

Reviewer #3 (Remarks to the Author):

The study by Scharzt et al. is aimed at demonstrating the potential of treatment with PMX205, a C5aR1 antagonist, in suppressing inflammatory glial gene expression and reversing some of cognitive symptoms observed in the Arctic mouse model of Alzheimer's disease. The authors performed an in-depth analysis of changes in gene expression in the microglia and other cell types by performing single cell and single nucleus analysis in the hippocampi isolated from Arctic mice (and wild types) treated with PMX205 and from those administered vehicle. The authors reported various differences in microglial and astrocytic gene expression. They then evaluated the histological presence of glial and astrocytic markers and also checked for the levels of neurofilament light. Finally, the short-term memory in the different mice groups was evaluated by performing the Y-maze test.

Major comments:

1.The work encompasses several different aspects of Alzheimer's disease in the Arctic model and provides an extremely rich source of information on the gene expression changes in this model and the rescue of some of these changes on administration of C5aR1 antagonist. However, the study lacks novelty in the sense that C5aR1 antagonism has been previously shown, by the same group, to be effective in alleviating some of the AD-specific changes in other mouse models of AD. The study and the data are undoubtedly an important resource that add value to AD literature.

2.It is confusing that the treatment paradigm is not consistent for all the different evaluations performed in the study. The treatment paradigm for single cell and single nucleus gene expression analysis is different from that used for qRT-PCR validation. The treatment paradigm for the mice used in the Y-maze test was not clearly mentioned in the methods. The early time point in the NfL testing experiment goes from 4.5-6.5 to 7 months of age; does this mean that for some mice the treatment started at 6.5 months and ended at 7 months and was only 15 days long?

3.A diagrammatic representation of the treatment paradigm for each of the experiments can be provided in the figures, which would make it easier to follow the time lines.

4.The rationale for performing both single cell and single nucleus analysis is not clear.

5.If the single cell analysis was performed on microglia isolated from the hippocampus, why was the enrichment of neuronal markers so robust, i.e. why are there more neuronal groups enriched in these samples? Was a validation test performed to ensure that the isolated microglial preparation was indeed enriched in microglia?

6.The analysis of microglial and astrocytic markers using histology could be performed in a stereotactic manner to ensure more rigorous evaluation of the markers in the entire hippocampus.

7.Why was the Y-maze test chosen? What kind of results do the authors expect to find in other memory tests like fear conditioning?

8. In figure 8A, the vehicle administered Arctic mouse group has only 3 samples, which is substantially smaller than that in the groups it is being compared with.

9. Diagrams (something like Venn diagrams) showing the genes in the different microglial and astrocytic groups that are changing in the Arc model and also those that are modified with the PMX205 treatment will be very helpful in understanding the data better. The text does not do justice to all the different parameters that are required to be kept track of by the reader.

Minor comments:

1. The y-axis labels in Fig 3 C and D do not display properly in the current version.

2. Fig 6 F, G, H, the abbreviation FA used in labeling the graph axis has not been defined.

Point-by-point response to the reviewers' comments, reproduced verbatim

REVIEWER COMMENTS

Reviewer #1 (Remarks to the Author):

Schartz et al. report the beneficial effects of treatment with a complement C5aR1 antagonist called PMX205 in the Arctic mutant APP mouse model of Alzheimer's disease. Interestingly, the drug prevented cognitive decline in female Arctic mice without reducing amyloid, suggesting that the mechanism behind the drug's efficacy may be due to alterations in the immune response and cell signaling. To address this question, the authors performed single cell and single nucleus RNAseq on hippocampal tissues comparing Arctic mice with and without PMX205 treatment. PMX205 treatment caused robust changes in microglial genes including downregulation of disease-associated microglia and upregulation of microglial genes associated with synaptic health and learning. PMX 205 also reduced expression of some reactive astrocyte genes. Overall, PMX205 promoted cell growth and repair and attenuated inflammatory pathways. The authors emphasize that this beneficial effect of PMX205 leaves the rest of the complement system intact to defend against pathogens. The methods and data/figures are clear and well-described.

Overall, this is a very well-written, highly significant paper in terms of both mechanisms of neuroprotection and support for targeting C5aR1 as a treatment for Alzheimer's disease. Importantly, this treatment strategy has implications for many other neurodegenerative diseases beyond Alzheimer's. Understanding the cellular pathways that underlie its beneficial effects may lead to additional therapeutic targets.

We are very pleased by your comment, thank you.

Comments:

1. Line 35 (Abstract): "C5aR1 inhibition promoted signaling pathways" – do you mean mRNA or protein? Please clarify here. *The pathways were identified by CellChat on mRNA data from single cell and single nucleus RNA-seq. We amended the abstract to include "mRNA-predicted signaling pathways."*
2. PMX205 reduced cerebral Aβ levels in Tg2576 mice but not in Arctic mice. Any idea of why there was this difference between mouse models? Do you think this was due to the timing of the intervention relative to disease stage or differences in the models? *We believe the difference in Aβ reduction between Tg2576 mice and Arctic mice is due to differences in the model. Our previous studies in the Arctic mice where C5aR1 was constitutively genetically ablated also showed very little to no effect on reducing amyloid plaques. The Arctic 48 APP mutation results in rapid amyloid fibril formation, a state that is more difficult to clear. As a result, it is a very aggressive amyloidogenic mouse model of AD, while the Tg2576 model has a much less temporally aggressive amyloid pathology accumulation. In both studies, involving the Tg2576 or Arctic mice treated with C5aR1 antagonist, we chose a time window specific for each model wherein plaque accumulation was evident but not yet peaked. Therefore, based on our previous results not only with PMX205 but also with a genetic ablation of C5aR1, we propose the differences observed between the Aβ levels are mainly due to differences in the mouse models. This clarification has been added to the Methods section describing*

the Arctic48 mouse (lines 488-490). Importantly, these differences have enabled us to distinguish many effects as being due to direct downstream consequences of C5a-C5aR1 signaling vs that due to the decrease in plaque load that is seen in the Tg2576 model.

3. Did PMX205 have any effect on CAA? This would be very interesting because CAA in human brain is associated with complement activation. *We did not detect CAA pathology in the Arctic mice (Hernandez, 2017, figure 3(1)). In addition, our previous results in the Tg2576 mice did not show any evidence of CAA and therefore, the effect of PMX205 on CAA could not be evaluated in these models. This would be an interesting question for future studies in models where CAA is evident.*

4. Line 61-62: Didn't Paul Morgan's group find C5b-9 colocalization with plaques? If so, their work should be cited here. *In his 2018 review (which we cited in the original line 61) Morgan showed a photomicrograph of MAC staining in AD brain and stated that the reports of C5b-9 were previously inconsistent. In a more recent paper (Carpanini et al., 2022 (2)) Webster,... Tenner,... Rogers, 1997 (3), is cited for the original report of C5b-9 in the AD brain, near—not on—plaques. C5b-9 is not nearly as dominant as the other complement components detected, particularly on plaques (one of a few reasons being there is little lipid membrane in the plaques to enable C5b-9 insertion). However, in this 2022 paper, Morgan and colleagues have demonstrated significant increases in C5b-9/MAC/TCC in total brain homogenates in the Suido AD mouse model (AppNL-G-F). A small but statistically significant loss of spine density was prevented by treatment with anti-C7 for two weeks demonstrating a role for C5b-9 in loss of spines within 50 μm of plaques, which they supported by a rescue of spine loss and synaptic density in C6 deficient 3xTg mice. This suggests that complement recognition and activation could lead to C5b-9 mediated induction of shedding of membrane patches that are then engulfed by microglia.*

We now include C5b-9 in now lines 62-63 and cite this important contribution.

5. Line 100-101: Why were WT and Arc mice treated with different doses of PMX205. Please explain the rationale for this. Was it due to weight differences between the mouse lines?

The differences in "achieved" doses depended on body weight and the amount each animal drank. All treated mice were given PMX205 in their drinking water at a concentration of 20 μg/ml. Wild type males weighed slightly more than Arc males and all females, but also consumed less volume, resulting in a lower dose in WT males relative to their body weight. This is now clarified in the results section (line 101 and 106) and remains also in the Methods section (lines 488-490).

6. In figures 2 and 4, it appears that PMX205 increases C3 expression in astrocytes. Typically, C3 upregulation is associated with an A1 (toxic) astrocyte response. How do the authors reconcile this seemingly pro-inflammatory effect of PMX205 in astrocytes? Please comment on this in the Discussion. *While often considering only complement cascade roles of inflammation and excessive synapse pruning (and thus being considered proinflammatory), a role for intracellular complement, coined the "complosome," has been described as a metabolic regulator for cells of the immune system ((4)). It has been hypothesized that as in specific immune cells, glial-derived C3 may be involved in intracellular signaling and/or metabolic reprogramming ((5-7)). For example, induction of astrocyte C3 under stress conditions (and thus DAA and originally A1) may very well have beneficial effects,*

particularly if it enhances metabolism and thus the astrocytic neuroprotective roles – which is seen in some astrocyte clusters identified here. This has been noted in the discussion at lines 390-396.

7. Any idea of why the DAM transcript changes were mostly seen with scRNAseq compared to snRNAseq? *Post-transcriptional processing is hypothesized to result in differences between the cytoplasmic RNAs and the nuclear RNA due to differences in stability or turnover rates, which is particularly evident in inflammatory cytokines.*

8. Did PMX205 have any effect on C5a levels? What about C5aR2 expression? *By ELISA, C5a levels in hippocampus or cortex were borderline or below the level of detection and not statistically different from WT when assayed in Arctic mice at 7 and 10 months of age (Carvalho et al., 2022)(8). This could be due to local generation and very rapid binding of C5a to upregulated C5aR1 and C5aR2 and internalization consistent with the decreases in C5a seen with disease progression in models of other neurological disorders (Lee et al., 2013) (9). As a result, we did not measure C5a concentrations in this study. In addition, Jain et al., (10) observed that PMX205 did not alter C3a nor C5a concentration measured using ELISA in colon explant supernatants of PMX205 treated mice compared to control. This would be expected as PMX205 binds the receptor and does not block cleavage of C5 and thus generation of C5a. This may be another advantage of PMX205 as it is hypothesized that C5a binding to C5aR2 (not blocked by PMX205), may have anti-inflammatory neuroprotective effects (11), as mentioned in the Discussion lines 454-459.*

In general, C5aR2 and C5aR1 are coexpressed (although some exceptions have been reported (12)) with C5aR1 usually more highly expressed than C5aR2. C5aR1 and C5aR2 are both upregulated at the mRNA level in the Arc mouse at 10 months. In earlier reports we showed by immunohistochemistry a marked increase in C5aR1 in microglia that are surrounding A β plaques AD mouse models vs those at a distance from plaques. While PMX205 reduced plaque load in these models (Tg2576 and 3xTg), C5aR1 levels on plaque associated microglia was not affected by PMX205 treatment (13). Here in the Arctic model both C5aR1 and C5aR2 show a slight decrease in mRNA with PMX205 but are not statistically significantly changed by PMX205 treatment (extracted from data obtained in this study) at the RNA level. We performed additional immunohistochemical staining and quantification of C5aR1 demonstrating no difference in reactivity with treatment with PMX205, which is consistent with the hypothesis that this early response to injury would not be altered by lack of signaling. These additional data are presented in Supplemental Figure S10, and now mentioned in the text, lines 294-295. However, antibody reagents for mouse C5aR2 are not well characterized and thus, we did not test for protein expression by IHC for C5aR2.

Reviewer #2 (Remarks to the Author):

Beneficial effects of C5aR1 antagonists have been shown before for therapeutic approaches of AD, ALS, spinal cord injury and other CNS disease, but the exact mechanisms were unknown. The authors' previous work, using a different AD model (TG2576) but the same C5aR1 antagonist (PMX205), showed a decrease in a unique microglia population responsible for synaptic pruning and increase of DAMP2 gene expressing microglia population associated with neuroprotection.

These present studies provide further data that pharmacologic inhibition of C5aR1 prevents deficit in

short-term memory of female mice in an aggressive model of AD. A more detailed characterization of microglia and astrocyte gene expression shows more microglia specific changes and activation of cell growth and repair associated pathways including TGF β , bone morphogenetic protein (BMP) and FGF activation in parallel with suppression of inflammatory pathways, such as PROS, Pecam1 and EPHA. While data presented in this paper might seem to be incremental, it is conclusive in demonstrating the beneficial effect of PMX205 in advanced amyloid pathology when substantial amyloid plaques are already present, highlighting its potential utility in AD intervention therapy.

We are pleased that this reviewer assessed our data as "...conclusive in demonstrating the beneficial effect of PMX205 in advanced amyloid pathology when substantial amyloid plaques are already present, highlighting its potential utility in AD intervention therapy."

The therapeutic effect in treated animals despite the absence of any impact on amyloid plaque load is interesting and is presented by the authors as an alteration of C5aR1-mediated reactivity/immune response to pre-existing amyloid plaques. While significant steps are still needed to understand the full extent of the complement activation pathway in human AD, this pharmacological target differs from the more direct plaque targeting drugs which have been investigated extensively and could prove to work synergistically with other candidate drugs. However, significant reductions in plaque accumulation in other models (3xTg and Tg2576) are still seen (up to 50%) with PMX205 and the failure of PMX205 to influence plaque load in Arctic mice was only briefly discussed.

*As mentioned in our response to Reviewer #1, point 2, our previous studies in the Arctic mice where C5aR1 was constitutively genetically ablated also showed very little to no effect on reducing amyloid plaques. The Arctic48 APP mutation results in rapid fibrillar formation, a state that is more difficult to clear. As a result, the Arctic mouse is a very aggressive amyloidogenic mouse model of AD, while the Tg2576 model has a much less aggressive temporal accumulation of amyloid pathology. This is an important point for our manuscript, as by using the Arctic mouse we are able to distinguish the direct effect of C5aR1 on the **cellular response to plaques** since the plaque load does not change +/- PMX205, whereas in our previous work with the Tg2576 mice the gene expression effects seen by inhibiting C5aR1 could be direct or the result of lowering the amount of plaques that result when inhibiting C5aR1 or both. The text has been revised in several places to emphasize this point.*

CellChat analysis led to the most intriguing data of this paper. According to this analysis, OPCs are the most prominent contributors to increased cell-cell signaling in AD that supports the highly adaptive and monitoring nature of these cells. It might be helpful to consider additional approaches to understand the significance of this information.

The paper is significant because this is the first study to provide a detailed gene expression profile analysis of multiple cell types during C5aR1 antagonist treatment in the aggressive arctic model of AD. However, despite multiple gene expression profiles being described, in the absence of deeper mechanistic studies, the overall mechanism and translational value are difficult to appreciate. Most of the data supports earlier reports of different mechanisms regarding neuroprotective effects.

We have extensively rewritten the Introduction, Results and Discussion to provide information on what was previously known about the intracellular molecular signaling pathways that result from C5a interaction with the C5aR1 (lines 69-74) and to clarify the contribution and impact of our new data to elucidating the mechanism and translational value. Specifically, we point to

- *the data that distinguish between C5aR1- dependent and C5aR1- independent responses to amyloid pathology evidenced by specific gene expression profiles determined at the single cell level.*
- *Importantly, since these are sc and sn RNA-seq derived data, they identify distinct microglia and astrocytes clusters present only when C5a-C5aR1 signaling was permitted, rather than a general modulation of all glia.*
- *Using the snRNA-seq to uncover predicted downstream consequences of inhibited C5aR1 signaling provides further and novel understanding of the possible mechanisms involving the intercellular interacting networks.*

*The potential translational value of these studies is that specifically targeting C5aR1 inhibits detrimental responses to injury while leaving the beneficial/neuroprotective responses to injury intact. In addition, importantly this can occur **with adult administration** of a drug after the onset of plaque deposition which more closely mimics what could occur in patients treated to inhibit C5aR1 to prevent or slow down AD progression.*

Furthermore, a C5aR1 antagonist has been FDA approved for peripheral disorders, indicating a safe therapeutic profile. (Also see Response to Reviewer #3, comment 1.).

Recent technical advances now enable future identification of spatial transcriptomic and proteomic information which will allow further understanding of the mechanism by which C5aR1 inhibition alters microglial cells and modifies AD progression.

Minor comment: Fig. 6A is missing “c” from CD11 *This format issue has been corrected.*

Fig. 5: SEMA7 should be SEMA7A. *In Figure 5, the CellChat output is describing pathways. This figure is labeling the SEMA7 pathway. In supplemental figure 9, the specific ligand, Sema7a, and its receptor Plxnc1 are described.*

Reviewer #3 (Remarks to the Author):

The study by Schartz et al. is aimed at demonstrating the potential of treatment with PMX205, a C5aR1 antagonist, in suppressing inflammatory glial gene expression and reversing some of cognitive symptoms observed in the Arctic mouse model of Alzheimer’s disease. The authors performed an in-depth analysis of changes in gene expression in the microglia and other cell types by performing single cell and single nucleus analysis in the hippocampi isolated from Arctic mice (and wild types) treated with PMX205 and from those administered vehicle. The authors reported various differences in microglial and astrocytic gene expression. They then evaluated the histological presence of glial and astrocytic markers and also checked for the levels of neurofilament light. Finally, the short-term memory in the different mice groups was evaluated by performing the Y-maze test.

Major comments:

1.The work encompasses several different aspects of Alzheimer’s disease in the Arctic model and provides an extremely rich source of information on the gene expression changes in this model and the rescue of some of these changes on administration of C5aR1 antagonist. However, the study lacks novelty in the sense that C5aR1 antagonism has been previously shown, by the same group, to be

effective in alleviating some of the AD-specific changes in other mouse models of AD. The study and the data are undoubtedly an important resource that add value to AD literature.

We apologize for the lack of focus related in the manuscript. The text and discussion have been rewritten to clarify these points. The new findings reported here include:

- *An important and novel aspect of the study presented here is that the plaque load does not change in the Arctic mouse upon inhibition of C5aR1. As mentioned above in our response to reviewer # 2, in our previous work with Tg2576 mice (14), mechanistically the effects seen by inhibiting C5aR1 could be merely the result of lowering the plaque load that occurs when inhibiting C5aR1 in those models. In this study, we are able to **distinguish** the direct effect of C5aR1 on the **cellular responses to plaques from the effects of plaque load** as plaque load was not reduced.*
- *Microglial and astrocyte responses to amyloid could also then be distinguished as C5aR1-dependent or C5aR1-independent responses, and defined by specific clusters of these cells. C5aR1-independent responses likely are due to responses to other innate danger recognition pathways (Eg. TLR signaling, Clec7a) as others have investigated. The value of C5aR1 inhibition is that excessive detrimental responses are suppressed, while protective (C5aR1-independent) glial responses are retained.*
- *While bulk transcriptome sequencing was reported in an earlier report in Arctic mice with a constitutive global genetic deletion of C5aR1, here RNAseq analysis (single cell and single nucleus) is assessed **after adult administration** of a drug after the onset of plaque deposition which more closely mimics what could occur in patients treated to prevent or slow AD progression. [While a conditional C5aR1 KO has been produced which would have enabled cell specific deletion of the receptor to further validate the direct response to C5a via C5aR1, we have invested considerable time demonstrating that in this mouse, while bone marrow derived myeloid cells expressed about 50% of the wild type levels of C5aR1 as published, brain induced C5aR1 is less than 10% that seen in mice (WT and Arc) without the floxed gene, suggesting an alteration/disruption in brain specific enhancers or accessibility to those elements as a result of the genetic engineering, rendering that mouse similar to the constitutive knock out in terms of brain C5aR1 expression. As a result, a new conditional model will need to be generated before cell specific changes shown here in response to a pharmacologic treatment can be corroborated with a time specific genetic deletion.]*
- *snRNAseq enables tracking of specifically induced or repressed subtypes of affected cell populations. That is, using the more recently matured snRNAseq technique, we are able to get insight into the effects of C5a-C5aR1 signaling on gene expression patterns of other cell types (astrocytes, neurons, oligodendrocytes, etc.) in the context of amyloid pathology, as well as identify novel transcriptionally distinct populations of a given cell type (such as the appearance of the novel subset/cluster of astrocytes (A7) that expresses neurotoxic genes).*
- *Furthermore, our data are consistent with microglial signaling to other cell types which then orchestrate additional steps in disease pathogenesis, which had been implied more recently with in vitro studies by Studer and colleagues (15).*
- *The fact that cell types other than microglia are affected provides insights on potential intercellular mechanisms. Further support for the multicellular response in AD etiology, some of which is C5aR1 dependent, was derived by applying our data to the CellChat program. Novel pathways for resilience are also suggested which can be further investigated in the future by us and by others.*
- *Finally, the 2022 PMX205 studies with Tg2576 mice were performed only in female mice (to maintain a consistent temporal pathology load as Tg2576 females accumulate amyloid plaques earlier than males). Here, in the Arctic mice, while there were some temporal differences in differential gene*

expression between males and females (Figure S3), both males and females showed similar directional changes.

2. It is confusing that the treatment paradigm is not consistent for all the different evaluations performed in the study. The treatment paradigm for single cell and single nucleus gene expression analysis is different from that used for qRT-PCR validation. The treatment paradigm for the mice used in the Y-maze test was not clearly mentioned in the methods. The early time point in the NFL testing experiment goes from 4.5-6.5 to 7 months of age; does this mean that for some mice the treatment started at 6.5 months and ended at 7 months and was only 15 days long?

The mice tested in the Y maze were the same mice as were analyzed in the RNAseq and IHC. Their treatment took place from 7.5 to 10 months of age. Following behavioral testing at ~9.5-10 months, plasma and brain samples were collected for IHC, RNAseq, and NFL analysis. A smaller cohort was treated from 4.5-6.5 months to 7 months of age (2, 6, and 12 weeks of treatment) for qPCR exploration of the effect of treating earlier and how quickly changes could be detected. We apologize for the lack of clarity. We have revised the text to improve understanding (lines 237-240, 249-251 and 325-328).

3. A diagrammatic representation of the treatment paradigm for each of the experiments can be provided in the figures, which would make it easier to follow the time lines.

We show a brief diagrammatic representation of the treatment paradigm in Figure 1A. To highlight this, we altered the description in the figure legend to “Diagrammatic representation of treatment paradigm for each of the experiments, including treatment from 4.5 to 7 months (top) and 7.5 to 10 months (bottom).”

4. The rationale for performing both single cell and single nucleus analysis is not clear.

We wanted to obtain transcriptomics data from single microglia and single astrocytes to predict the interactions between these two cell types. The protocol for single cell isolation was optimized for microglia, but we were unable to capture a reactive astrocyte population with similar methods. On the other hand, single nucleus isolation is able to capture neurons, astrocytes, and other cell types. However, due to the low proportion of microglia naturally occurring in the brain (~10%), we decided to use single cell isolation to capture a microglia-enriched population and single nucleus to capture all other cell types.

5. If the single cell analysis was performed on microglia isolated from the hippocampus, why was the enrichment of neuronal markers so robust, i.e. why are there more neuronal groups enriched in these samples? Was a validation test performed to ensure that the isolated microglial preparation was indeed enriched in microglia?

As the reviewer rightly states, single cell analysis was performed on microglial cells isolated from the hippocampus. However, the single nucleus sequencing was done on separate preparations from other hippocampi. It is those snRNA-seq data of the hippocampus that are enriched in neuronal groups/markers, not the single cell microglia samples. Indeed, all Seurat clusters from the combined sn and sc sequencing were analyzed via the top 100 genes to assign to a cell type. Clusters 3, 4, 6, 9, 11, 12, and 14 were identified as either ependymal, endothelial, pericytes, astrocytes, or “other” based on gene expression (Figure 1 and 2). Violin plots on the reclustered “microglia” showed expression of microglial markers in all clusters except #11 which was discarded from subsequent analyses (Fig. S2B).

Only those clusters verified as having CSFR1, Tmem119, Cx3CR1 and P2yr12 were further analyzed as presented in Figure 3. Violin plots were also used to characterize the astrocyte clusters (Fig. S2C).

6. The analysis of microglial and astrocytic markers using histology could be performed in a stereotactic manner to ensure more rigorous evaluation of the markers in the entire hippocampus.

We thank the reviewer for this comment. The reviewer is right, and stereology measurements would be suitable to evaluate the entire hippocampus. However, given our initial assessment of these markers and limited availability of a stereology microscope, as an alternative, we now performed additional IHC for microglial (Iba1, CD11b and CD11c) and astroglial (s100a6, LCN2 and GFAP) markers in 3-4 sections per mouse with a large group of mice (n=3-8 mice/genotype/treatment). We then obtained 10x images with a Zeiss Axioscan Z1 digital slide scanner in order to image the entire hippocampus (instead of the confocal images of smaller portions of the hippocampus shown in the original manuscript), resulting in a total of 9-24 whole hippocampal images per group. Figure 6 has now been revised:

Fig 6 – Iba1 quantification: each dot represents the average of 3-4 sections per mouse and: WT Veh (N=6), WT PMX (N=5), Arc Veh (N=7) and Arc PMX (N=7).

Fig 6 – Cd11b and Cd11c quantification: each dot represents the average of 3-4 sections per mouse and: WT Veh (N=6), WT PMX (N=6), Arc Veh (N=7-8) and Arc PMX (N=8)

Fig 6 – S100a6, LCN2 and GFAP quantification: each dot represents the average of 3-4 sections per mouse and: WT (N=3), WT PMX (N=3), Arc Veh (N=7-8), Arc PMX (N=5).

Therefore, we believe that our data now represents a rigorous evaluation of the markers in the entire hippocampus.

7. Why was the Y-maze test chosen? What kind of results do the authors expect to find in other memory tests like fear conditioning?

In this cohort, object location memory deficits or contextual fear memory deficits were not seen in the Arctic mice at 10 months of age. The Y-maze test had previously yielded reliable results at this age, and thus was chosen to test this treatment focusing on hippocampal spatial cognition.

8. In figure 8A, the vehicle administered Arctic mouse group has only 3 samples, which is substantially smaller than that in the groups it is being compared with.

To clarify the results, we have switched the positions of Figures 8A and 8B and the associated text. The data in Figure 8B (Formerly 8A) was a smaller group of animals treated at a younger age and perfused at 7 mo to probe for effects of earlier treatment on neurodegeneration. Figure 8A (formerly 8B) includes samples from mice that were also included in behavior, and thus consists of a greater sample size (in order to be adequately powered for behavior). Indeed, it did appear that earlier treatment may have had a greater effect.

9. Diagrams (something like Venn diagrams) showing the genes in the different microglial and astrocytic groups that are changing in the Arc model and also those that are modified with the

PMX205 treatment will be very helpful in understanding the data better. The text does not do justice to all the different parameters that are required to be kept track of by the reader.

We have added a supplemental figure (Figure S13) which demonstrates the cell clusters that are modified (or not) by PMX205 treatment in Arctic microglia (A) and astrocytes (B). All of these clusters were increased in the Arctic compared to WT but responded differently to PMX205 treatment.

Minor comments:

1. The y-axis labels in Fig 3 C and D do not display properly in the current version. *We have fixed this formatting issue.*
2. Fig 6 F, G, H, the abbreviation FA used in labeling the graph axis has not been defined. *The definition has been added to the figure legends.*

We appreciate the thoughtful comments of the reviewers which have guided our revisions to produce a more clear and concise report. We hope the revisions will be acceptable for the reviewers, and that our manuscript will now be accepted for publication in Nature Communications.

Sincerely,

Andrea J Tenner, PhD

Distinguished Professor

University of California, Irvine

1. Hernandez MX, Jiang S, Cole TA, Chu S-H, Fonseca MI, Fang MJ, et al. Prevention of C5aR1 signaling delays microglial inflammatory polarization, favors clearance pathways and suppresses cognitive loss. *Molecular Neurodegeneration*. 2017;12(1):66.
2. Carpanini SM, Torvell M, Bevan RJ, Byrne RAJ, Daskoulidou N, Saito T, et al. Terminal complement pathway activation drives synaptic loss in Alzheimer's disease models. *Acta Neuropathol Commun*. 2022;10(1):99.
3. Webster S, Lue LF, Brachova L, Tenner AJ, McGeer PL, Terai K, et al. Molecular and cellular characterization of the membrane attack complex, C5b-9, in Alzheimer's disease. *NeurobiolAging*. 1997;18(4):415-21.
4. West EE, Kemper C. Complosome - the intracellular complement system. *Nat Rev Nephrol*. 2023;19(7):426-39.
5. Kunz N, Kemper C. Complement Has Brains-Do Intracellular Complement and Immunometabolism Cooperate in Tissue Homeostasis and Behavior? *Front Immunol*. 2021;12:629986.
6. Kolev M, Kemper C. Keeping It All Going-Complement Meets Metabolism. *Front Immunol*. 2017;8:1.
7. Hess C, Kemper C. Complement-Mediated Regulation of Metabolism and Basic Cellular Processes. *Immunity*. 2016;45(2):240-54.
8. Carvalho K, Schartz ND, Balderrama-Gutierrez G, Liang HY, Chu SH, Selvan P, et al. Modulation of C5a-C5aR1 signaling alters the dynamics of AD progression. *J Neuroinflammation*. 2022;19(1):178.

9. Lee JD, Kamaruzaman NA, Fung JN, Taylor SM, Turner BJ, Atkin JD, et al. Dysregulation of the complement cascade in the hSOD1G93A transgenic mouse model of amyotrophic lateral sclerosis. *J Neuroinflammation*. 2013;10:119.
10. Jain U, Woodruff TM, Stadnyk AW. The C5a receptor antagonist PMX205 ameliorates experimentally induced colitis associated with increased IL-4 and IL-10. *Br J Pharmacol*. 2013;168(2):488-501.
11. Biggins PJC, Brennan FH, Taylor SM, Woodruff TM, Ruitenbergh MJ. The Alternative Receptor for Complement Component 5a, C5aR2, Conveys Neuroprotection in Traumatic Spinal Cord Injury. *J Neurotrauma*. 2017.
12. Desai JV, Kumar D, Freiwald T, Chauss D, Johnson MD, Abers MS, et al. C5a-licensed phagocytes drive sterilizing immunity during systemic fungal infection. *Cell*. 2023;186(13):2802-22 e22.
13. Ager RR, Fonseca MI, Chu SH, Sanderson SD, Taylor SM, Woodruff TM, et al. Microglial C5aR (CD88) expression correlates with amyloid-beta deposition in murine models of Alzheimer's disease. *J Neurochem*. 2010;113(2):389-401.
14. Gomez-Arboledas A, Carvalho K, Balderrama-Gutierrez G, Chu SH, Liang HY, Schartz ND, et al. C5aR1 antagonism alters microglial polarization and mitigates disease progression in a mouse model of Alzheimer's disease. *Acta Neuropathol Commun*. 2022;10(1):116.
15. Guttikonda SR, Sikkema L, Tchieu J, Saurat N, Walsh RM, Harschnitz O, et al. Fully defined human pluripotent stem cell-derived microglia and tri-culture system model C3 production in Alzheimer's disease. *Nat Neurosci*. 2021.

REVIEWERS' COMMENTS

Reviewer #1 (Remarks to the Author):

The authors have done a tremendous job at addressing completely and consistently all concerns and queries regarding their submitted manuscript. They have added new data, provided more detailed information, and have updated their Discussion section to be more relevant and inclusive of limitations of interpretation. As a result, the paper is much improved. This is an important paper that will help the field better understand how cellular immune signaling in the brain can influence brain function. The therapy they tested provides evidence that blocking C5a/C5aR1 signaling in amyloid mouse models of Alzheimer's disease may be neuroprotective. The authors are to be commended for their high quality studies and impactful data. I have no further concerns.

Reviewer #2 (Remarks to the Author):

The authors addressed the concerns adequately. The manuscript convincingly shows that targeting C5aR1 inhibits detrimental responses in AD-induced inflammation while leaving the beneficial responses intact.

Reviewer #3 (Remarks to the Author):

Thank you for addressing the concerns. I have nothing more to add other than saying that this is a thorough study that significantly improves our understanding of AD.

Point-by-point response to the reviewers' comments, reproduced verbatim

REVIEWER COMMENTS

Reviewer #1 (Remarks to the Author):

The authors have done a tremendous job at addressing completely and consistently all concerns and queries regarding their submitted manuscript. They have added new data, provided more detailed information, and have updated their Discussion section to be more relevant and inclusive of limitations of interpretation. As a result, the paper is much improved. This is an important paper that will help the field better understand how cellular immune signaling in the brain can influence brain function. The therapy they tested provides evidence that blocking C5a/C5aR1 signaling in amyloid mouse models of Alzheimer's disease may be neuroprotective. The authors are to be commended for their high quality studies and impactful data. I have no further concerns.

We are very pleased by your comments, thank you.

Reviewer #2 (Remarks to the Author):

The authors addressed the concerns adequately. The manuscript convincingly shows that targeting C5aR1 inhibits detrimental responses in AD-induced inflammation while leaving the beneficial responses intact.

Thank you.

Reviewer #3 (Remarks to the Author):

Thank you for addressing the concerns. I have nothing more to add other than saying that this is a thorough study that significantly improves our understanding of AD.

Thank you. Our goal is to promote progress toward mitigating this disease.

We thank the reviewers for their thoughtful comments which have led to a more clear and impactful report.